# Data-driven CAD-CAM vs traditional total contact custom insoles: A novel quantitative-statistical framework for the evaluation of insoles offloading performance in diabetic foot

**Moreno D'Amico**[1,2]\*, **Edyta Kinel**[3], **Piero Roncoletta**[1], **Andrea Gnaldi**[4], **Celeste Ceppitelli**[5], **Federico Belli**[6], **Giuseppe Murdolo**[5], **Cristiana Vermigli**[5]

**1** SMART Lab (Skeleton Movement Analysis and Advanced Rehabilitation Technologies)—Bioengineering & Biomedicine Company Srl, Pescara, Italy, **2** Department of Neuroscience, Imaging and Clinical Sciences University G. D'Annunzio, Chieti-Pescara, Italy, **3** Department of Rehabilitation, Chair of Rehabilitation and Physiotherapy, University of Medical Sciences, Poznan, Poland, **4** Guantificio Altotiberino Ecosanit Calzature Snc, Anghiari, Italy, **5** Department of Medicine, Unit of Endocrinology and Metabolism, S. Maria della Misericordia, Perugia Hospital, Perugia, Italy, **6** Independent Researcher, Perugia, Italy

\* damicomoreno@gmail.com

## Abstract

### Background

Elevated plantar pressures represent a significant risk factor for neuropathic diabetic foot (NDF) ulceration. Foot offloading, through custom-made insoles, is essential for prevention and healing of NDF ulcerations. Objective quantitative evaluation to design custom-made insoles is not a standard method. Aims: 1) to develop a novel quantitative-statistical framework (QSF) for the evaluation and design of the insoles' offloading performance through in-shoe pressure measurement; 2) to compare the pressure-relieving efficiency of traditional shape-based total contact customised insoles (TCCI) with a novel CAD-CAM approach by the QSF.

### Methods

We recruited 30 neuropathic diabetic patients in cross-sectional study design. The risk-regions of interest (R-ROIs) and their areas with in-shoe peak pressure statistically $\geq$200kPa were identified for each patients' foot as determined on the average of peak pressure maps ascertained per each stance phase. Repeated measures Friedman test compared R-ROIs' areas in three different walking condition: flat insole (FI); TCCI and CAD-CAM insoles.

### Results

As compared with FI (20.6±12.9 cm$^2$), both the TCCI (7±8.7 cm$^2$) and the CAD-CAM (5.5 ±7.3 cm$^2$) approaches provided a reduction of R-ROIs mean areas (p<0.0001). The CAD-CAM approach performed better than the TCCI with a mean pressure reduction of 37.3 kPa (15.6%) vs FI.

**Data Availability Statement:** All relevant data are within the manuscript and its Supporting Information files.

**Funding:** The author(s) received no specific funding for this work.

**Competing interests:** I have read the journal's policy and the authors of this manuscript have the following competing interests: Dr. D'Amico and Dr. Roncoletta own shares of the Bioengineering & Biomedicine Company Srl. This does not alter their adherence to Plos One policies on sharing data and materials. Dr Gnaldi is an employee of Guantificio Altotiberino Ecosanit Calzature Snc. This does not alter their adherence to Plos One policies on sharing data and materials Dr. Kinel, Dr. Vermigli, Dr. Ceppitelli, Dr. Murdolo, Dr. Belli have declared that no competing interests exist.

## Conclusions

The CAD-CAM strategy achieves better offloading performance than the traditional shape-only based approach. The introduced QSF provides a more rigorous method to the direct 200kPa cut-off approach outlined in the literature. It provides a statistically sound methodology to evaluate the offloading insoles design and subsequent monitoring steps. QSF allows the analysis of the whole foot's plantar surface, independently from a predetermined anatomical identification/masking. QSF can provide a detailed description about how and where custom-made insole redistributes the underfoot pressure respect to the FI. Thus, its usefulness extends to the design step, helping to guide the modifications necessary to achieve optimal offloading insole performances.

## Introduction

Neuropathic diabetic foot (NDF) ulceration, one of the most frequently recognised complications in diabetes mellitus, represents a health concern [1,2]. Besides the increased risk of foot infection and limb amputation [3], patients with NDF ulceration show increased mortality rate as compared with diabetic subjects without a foot ulcer [4], underscoring the paramount importance of prevention [5].

Retrospective and prospective studies showed that increased mechanical load on foot, in particular, elevated plantar pressures during walking is a causative factor in the development of plantar ulcers in diabetic patients. Ulceration is often a precursor of lower extremity amputation [6]. Interventions aimed at reducing these high plantar pressures are referred to as "offloading", one of the cornerstones of treatment for either preventing/treating or healing plantar NDF ulceration [7].

Suitably designed insoles are proven to reduce forefoot plantar pressures and risk of re-ulceration. They act primarily by redistributing and relieving high plantar pressure levels [2,3,7–15], but poor adherence is a crucial barrier to clinical success, and it is increasingly being recognised and considered fundamental in clinical guidelines [2,5,16]. Footwears also play an essential role in such a crucial objective to reduce foot pressures as they work in conjunction with the custom-made insole. Such a role appears particularly evident when the use of rigid rocker-configured outsoles is considered useful as an additional intervention.

As strongly underlined in van Netten et al. [15], several healthcare disciplines may be involved in the provision of footwear and insoles for people with diabetes. Experts from different fields may use different vocabulary, introducing possible confusion and misunderstanding. So, a common vocabulary is essential for clear communication. To this aim, we refer to the definitions proposed in [15], where the terms "custom-made insoles" and "custom-made in-shoe orthosis/orthotic" are equivalent and in a such a way they will be used throughout this paper. Despite the offloading goal, objective quantitative evaluation to control the pressure-relieving properties of custom-made insoles is still not a standard method in NDF ulceration prevention/healing practice [5]. Foot orthotics are mostly evaluated based on clinical experience and a trial-and-error approach [11]. Conversely, studies introducing the routinely use of quantitative plantar pressure measurement-driven directions for effectively offloading insoles are increasingly presented in the literature [2,3,7,5,8–14,17]. Several studies have investigated the offloading effects either induced by elevating pressures in areas of lower pressure (e.g. medial arch support and metatarsal pads or bar) [3,9–11,18–20] or by the use of soft materials

in areas of high pressure (e.g. forefoot cushion) [21] or the combination of the two [12,17]. In these studies, various optimisation strategies have been presented to find the "optimal offloading design". They are based on the 3D shape, and plantar pressure distributions information either used singularly (for total contact insoles) [13,22,23] or in some combined way [3,8,9,11,17,24,25]. Plantar pressure can be redistributed but not eradicated and reducing stress at one location may simply displace the risk of ulceration to a different area of the foot [12]. Such an occurrence must be avoided using quantitative control. In cases of re-ulceration, reducing plantar pressures to below 200 kPa has been advocated [10,11,26]. An equivalent threshold does not exist for first ulceration, pre-first ulceration plantar tissue is likely less vulnerable to external loads [27]. In any case, in the absence of other proven thresholds, the 200kPa limit is taken as a standard reference. Although these custom-made insole-design approaches significantly reduce plantar foot pressure, their effects have been studied in small groups of participants and outcomes are variable across patients [17,18,20,22], even if controlled clinical trials are increasingly presented in the literature [2,6,15,28]. To guarantee the best as possible offloading performance along time, a well-established periodic systematic control to verify and eventually modify/correct insoles' characteristics has to be set [3,11]. The control period, i.e. the rate of evaluation needed is to be individually determined given each patient's specific pathological level and anatomical/loading pattern characteristics, the distinctive wearing of material used in insole fabrication etc. Despite such increasing claim on the use of available measuring technology for both under-foot pressure and 3D shape quantifications, the debate is open about the real accessibility of such technologies into the routine clinical environment [5]. The strategies to design the optimal offloading insoles have to be fully standardised [25,29]. In the absence of such standardisation, outcomes for digital quantitative data-driven approaches can be debatable in comparison of traditional foot orthoses supply chains [13]. A further consideration is about gait intrinsic variability. At the authors' knowledge, no offloading insole design study presented in the literature considers such a topic. The gait is a cyclic sequence of stance and flight phases during the forward moving of the body. Despite its cyclical nature, the ways in which the support phases follow one another always present minimal differences from each other, either in terms of temporal duration or in terms of foot-floor interaction loads. Such variations are present both in physiological and pathological gait. Such variability reflects naturally on the underfoot pressure. So, variability must be considered when the underfoot pressure measurements are used in custom-made insole design, and in their outcome performances monitoring. These intrinsic gait-related variations signify the importance of approaching the study of its characteristics from a statistical perspective. Given the open debate on these themes, the aim of this study was twofold. Firstly, we wanted to build a sound quantitative-statistical framework (QSF) to analyse in-shoe pressure measurements, taking into account gait variability, to propose a way to standardise the evaluation of offloading insoles' design and subsequent monitoring steps. Secondly, we applied such QSF to compare the outcomes of traditional shape-based total contact foot orthotics design with a novel CAD-CAM approach based on a mixed 3D shape and pressure measurement information for the placement of offloading features.

## Materials and methods

### Participants

Thirty neuropathic diabetic patients at risk for plantar foot ulceration (16 males, 14 females, mean age 67.9 and standard deviation (SD) (±11.6) years, mean body mass index (BMI) 25.9 (±3.7) were selected to participate in the study, from a pool of 220 patients screened in the Diabetic Foot Centre at the Department of Endocrinology and Metabolism of Perugia University

Hospital. Average time since diabetes onset was 23.5 (±13.5) years. Seven patients had diabetes type 1; twenty-three patients had diabetes type 2. The patients were eligible for the study if they met the following inclusion criteria: 1) age >18 years; 2) a diagnosis of diabetes mellitus type 1 or type 2; 3) Peripheral Neuropathy 4) Foot ulcer risk classification classes scored Grade 1 to Grade 3 using the IWGDF Risk Stratification System [6]; 5) a measured in-shoe plantar pressure >200 kPa; 6) absence of active foot lesions; 7) ability to walk a minimum of 30 m unaided; and, 8) acceptable metabolic control shown as HbA1c <8.5%. Patients were excluded if they presented with: 1) motor/sensory impairment not allowing unassisted locomotion; 2) body mass index (BMI) $\geq$33 kg/m$^2$; 3) severe fixed midfoot or rearfoot deformity such as that associated with Charcot arthropathy; 4) bilateral or multiple ulcers on a single foot; 5) acute vascular problems; 6) psychiatric disorders interfering with patient compliance; 7) a history of lower limb amputation.

Informed written consent was obtained from all volunteers before they participated in the study, which has been approved by the Ethical Committee of Umbria Region (CEAS N. 2241/14) and carried out according to Declaration of Helsinki principles.

## Neuropathy and risk classification

All patients had peripheral sensory neuropathy, indicated by a loss of protective sensation in the foot through the inability to sense the 10 g Semmes-Weinstein monofilament in at least four plantar foot sites tested [30]. Each patient was administered with the Diabetic Neuropathy Index (DNI), a simple eight-point clinical examination [31], to increase the sensitivity of peripheral neuropathy detection [32]. Patients with total DNI >2 over a total of 8 points were classified as neuropathic [31]. Final classification resulted in 18 subjects classified in risk class 1, 3 in risk class 2 and 9 subjects in risk class 3 [6].

Table 1 summarises the risk characteristics of the participants.

## Study design

The experimental setup was configured to perform a double-blinded two measurements sessions. In the first session, the subjects were provided with special sandals (sandal with heel containment buttress and 3 velcro fastening model: ELECTRA- Guantificio Altotiberino

**Table 1. Risk related characteristics of the participants.**

| Participant Characteristics | Values/Percentages | |
|---|---|---|
| N. Patients | 30 | |
| HbA1c | 7.40±0.87 | |
| History of Foot Ulcer | 26.6% (8/30) | |
| Risk Class 1 | 60.0% (18/30) | |
| Risk Class 2 | 10.0% (3/30) | |
| Risk Class 3 | 30.0% (9/30) | |
| Ankle-Brachial Pressure Index | Right 1.13±0.2 | Left 1,14±0.2 |
| Foot deformities | 53.3% (16/30) | |
| Bony prominence | 46.6% (14/30) | |
| Limited Joint Mobility | 16.6% (5/30) | |
| Hyperkeratosis | 100% (30/30) | |
| Foot small muscle atrophy | 20.0% (6/30) | |
| Onychodystrophy | 66.6% (20/30) | |
| Previous use of Foot Orthoses | 56.6% (17/30) | |

ECOSANIT Calzature Snc, Anghiari, Italy) with a flat insole (FI). Pictures, blueprints and foam impressions for each foot were also performed (Fig 1A and 1D). In the second session, the FI was removed, and the same sandals were able to accommodate the custom-made insoles. Double-blinded design, in this research, is related to the condition that neither the patients nor the operators who performed data recording were aware of which kind of custom-made insoles were under test. To keep under control possible biasing sources in the statistical analysis, we chose to adopt a single model of sandals for every patient and each measurement session. Indeed, we wanted to reduce the confounding effects arising from the use of the subject's different shoes. Secondly, we wanted to avoid any interference in the underfoot pressure measurements deriving by eventual shoe-upper unfit concerning different shapes or deformities of the patient's foot. In fact, a sandal does not present any shoe-upper that could interact/conflict with the dorsal aspect of the foot so potentially inducing sources of unexpected underfoot pressure.

## Foot orthoses

All the FI and the custom-made foot orthoses were fabricated using 12.7-mm-thick [33–35] ethylene-vinyl acetate foam (EVA with shore 40) with 1.5 mm Professional Protective Technology (PPT) top cover [9,24,36]. The FI is a simple 12.7-mm+1.5mm PPT thick neutral insole provided with the sandals (Guantificio Altotiberino ECOSANIT Calzature Snc, Anghiari, Italy). Foam impressions were made separately for each foot. Then, blueprints images of the feet on a millimetric grid were recorded in order to identify the foot shape and mark, with a red pen, all at-risk locations (i.e. callosities; sites of previous ulcers Fig 2).

The foam impressions were scanned using a 3D scanner (Guantificio Altotiberino ECOSANIT Calzature Snc, Anghiari, Italy) and then immediately sent to an orthotic supply company for the fabrication of total contact custom insoles (TCCI) [22,37,38] manufactured with the thickness, materials, and top cover defined above. TCCIs were fabricated using individual positive plaster moulds, based on the received negative foam impressions.

The 3D scans were sent to a different orthotic supply company for the subsequent CAD-CAM processing to prepare the custom milled insoles (CAD-CAM_CMI). Fig 1 depicts the fabrication processes for both the considered approaches.

Fig 2 shows an example of the recorded blueprints with related at-risk locations (i.e. callosities and sites of previous ulcers).

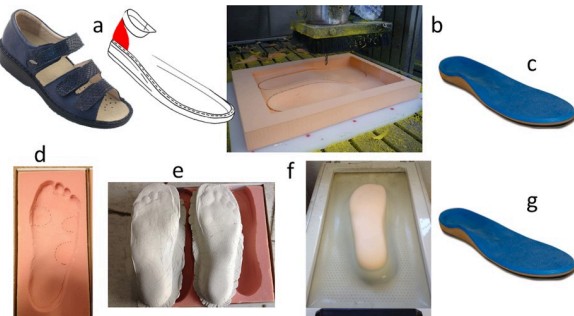

**Fig 1. Measuring and custom-made insoles fabrication process.** Specific interchangeable insole sandals used in the experimental setup (a). CAD-CAM milling process (b). CAD-CAM_CMI hand-finished by the application of a PPT top cover (c). Foam box impression with marked at-risk areas (d). Individual positive plaster moulds (e). Vacuum formation (f). Manufactured TCCI hand-finished by the application of a PPT top cover (g). Reprinted under a CC BY license, with permission from Guantificio Altotiberino ECOSANIT Calzature Snc, Anghiari, Italy, original copyright 2020.

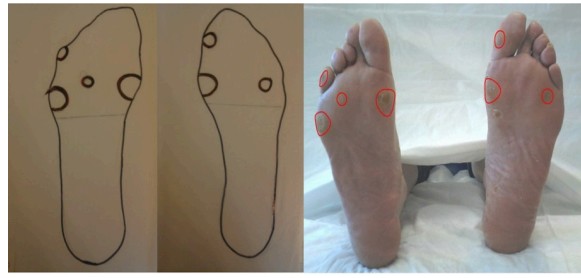

**Fig 2. Patient's blueprint with related at-risk locations.** Example of the recorded blueprints with related at-risk locations (i.e. callosities and sites of previous ulcers).

A foot orthotist designed the CAD-CAM_CMI foot orthoses by the ECOPLAN (Guantificio Altotiberino ECOSANIT Calzature Snc, Anghiari, Italy) software (A 3D CAD-CAM software specially developed for footwear design), according to the subject's foot scan and mean peak pressure maps (MPPM). Such a method is similar to that presented by Owings et al. [9]. The main substantial differences, in the actual CAD-CAM_CMI design approach, are related to the use of baropodometric insoles to determine the mean pressure maps driving the insole design and to the statistically rigorously determined number of necessary steps to be averaged (see below). MPPM plantar pressure data (see next Data Analysis section), which were integrated into its algorithms for the manufacture of the custom insoles, were also provided. By combining shape and pressure data, the ECOPLAN (Guantificio Altotiberino ECOSANIT Calzature Snc, Anghiari, Italy) software allowed the superimposition of the shape and pressure contours onto an outline of the planned insole perimeter (Fig 3). The primary offloading technique used for the custom milled insoles was the removal of material under the high-pressure area. An automated design algorithm identifies the high-pressure contours along which the 3D scan shape is modified, creating a proportional pressure deepening, in the area of the insole underneath

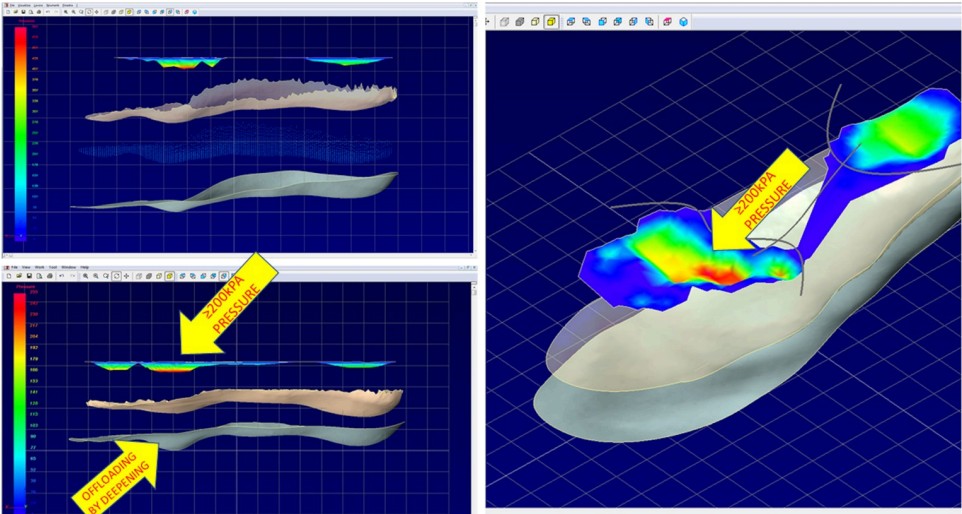

**Fig 3. CAD-CAM Software for foot orthoses design.** The ECOPLAN software combines shape and pressure data allowing the superimposition of the shape and pressure contours onto an outline of the intended insole perimeter. The offloading technique used is the removal of material under the high-pressure area. An automated design algorithm identifies the high-pressure contours along which the 3D scan shape is modified, creating a deepening area of the insole underneath regions of excessive local pressure (>200 kPa). Reprinted under a CC BY license, with permission from Guantificio Altotiberino ECOSANIT Calzature Snc, Anghiari, Italy, original copyright 2020.

regions of excessive local pressure (>200 kPa) [10,11,26]. The automated design algorithm details are the intellectual property of Guantificio Altotiberino ECOSANIT Calzature Snc, Anghiari, Italy, and they were not disclosed to the authors being not strictly necessary for the present study. The resulting insole was manufactured on a computer numerical control milling machine and hand-finished by the application of a 1.5mm PPT top cover [36] (Fig 1G).

Minimum thickness limits of 12.7mm were used in both manufacturing procedures [33–35]. After the two pairs of custom insoles per subject were ready at the research laboratory (average needed fabrication time for both kind of orthoses around 7–10 days), the patients returned for the second experimental session involving the measurement of in-shoe plantar pressures with each type of custom insole. The fabrication time was kept as short as possible in order to avoid confounding effects due to possible changes in patient status. Sandals were used only during the test sessions.

In the second experimental session, one medical doctor (author CV) was in charge to deliver the insoles to the patients and to keep track of the insoles type used in the test. According to the double-blind design, the coatings of both custom-made insoles were visually indistinguishable, so the patient and the technician in charge of the data acquisition ignored the currently used insole type. During the second experimental session, the two testing conditions (TCCI and CAD-CAM_CMI) were randomly presented throughout the recording to each of the participants to avoid confounding effects due to eventual fatiguing process. Subjects performed multiple passes along the 15-m walkway to collect data from at least 24 mid-walk foot-contacts for each foot for each condition. Once an appropriate number of steps had been collected, the current insole-shoe condition was removed and replaced with the next one until data from all conditions were gathered.

## In-shoe dynamic plantar pressure measurement

In-shoe dynamic plantar pressure was measured using the PEDAR-X system (Novel GmbH, Munich, Germany). The device comprises flexible 2 mm thick insoles with a matrix of 99 capacitance-based sensors (presenting different sizes depending on their relative location) each sampling at 50 Hz, which were placed in the shoes between the sock and insole. In-shoe plantar pressure was measured within a range from 20 to 600 kPa. Before pressure assessment, a 'zero-calibration' was performed by unloading each measurement insole while the patient wore experimental sandals. In-shoe plantar pressure was assessed while walking in multiple trials along a 15-m walkway in a laboratory setting. Before data collection, subjects established their average self-selected comfortable walking speeds, measured using a photocell system, by walking in their sandals along the 15-m path. The gait pace was also noted and used providing the subjects with a synching pace acoustic signal as a reference to help and simplify the control of walking speed during the first and second measurement sessions. In this way, by using both the photocell system and the acoustic signal, it resulted relatively easy to limit speed variations in a ±5% range during the data acquisition sessions. When accidentally a trial was exceeding such ±5% range it was removed from storage, and a further valid trial was recorded. To control the data collection via the PEDAR-X system directly, we used a customised real-time acquisition software (Bioengineering & Biomedicine Company Srl, Pescara, Italy). Following each acquired gait trial, this software was able to store the exported data (EMASCII format) from PEDAR-X software (Novel GmbH, Munich, Germany) for subsequent elaborations. During EMASCII data export, the pressure maps provided by the matrix, which consist of a grid of ninety-nine capacitance-based sensors of variable dimensions, are interpolated and re-sampled into a rectangular matrix of regular dimensions, in which every single element is represented by a square of 5 mm x 5 mm.

## Data analysis

The gait is a cyclic sequence of stances phases and flight phases during the transfer forward of the body. Despite its cyclical nature, how support phases follow one another always present minimal differences from each other either in terms of temporal duration or in terms of foot-floor interaction loads. Such intrinsic variability connected to gait, both normal and pathological one, implies the necessity to approach the study of its characteristics from a statistical point of view. That is, by defining a rigorous averaging process to extract mean behaviours and their associated variability from which to derive clinically relevant parameters with a statistical significance.

In-shoe pressure data were analysed using BIG (Baropodometry Integrated in Gait -Bioengineering & Biomedicine Company Srl, Pescara, Italy) software. For each condition, the peak pressure maps of all collected steps were averaged for each foot. Each peak pressure maps represent the grids obtained with the highest plantar pressure values recorded in every single cell during each stance phase. The walking trials with FI collected during the initial experimental session were used to identify initial high-pressure at-risk areas. Once at least 24 stance phases were collected in each foot, the mean and SD peak pressure maps were computed by assigning, to each cell, the averaged value with its SD calculated from the whole sequence of at least 24 peak pressure maps, respectively [39].

Any single cell of the obtained mean peak pressure map (i.e. each 5mm x 5mm grid element) that had a statistically significant peak pressure greater-equal than 200 kPa [10,11,26] was considered to compose an element of a risk-region of interest (R-ROI). We underline that such statistical R-ROIs determination permits to manage for the intrinsic gait variability considering both the mean value and its related SD. In such a way, the drawback of a simple direct threshold comparison performed on a mean peak frame map is overcome by making the determination of R-ROIs position and extent a theoretically more rigorous process. Indeed, a series of cells having average values fairly below the 200 kPa threshold but with high SD (let us say μ = 170kPa±30kPa where SD = 30kPa) would definitely not be considered by the direct threshold method. Conversely, if the variability of gait were found to be high producing a high SD (as in the numerical example), the one-sample t-test (included in the QSF) would likely consider this series of cells not statistically different from the 200 kPa threshold. Thus, the potential risk associated with this series of cells is not neglected by including them in the R-ROIs. For these reasons, the QSF approach is more conservative considering the statistical risk.

Each position of these cells was marked in the full grid, which, in turn, represents the whole contact area of the foot. The R-ROIs were thus represented by areas of one or few cells or by broader clusters of cells. Worth to note that, in this way, the determination of the R-ROIs position and size along the underfoot surface is connected only to their statistical over-threshold pressure values. For such a reason we did not give any a priori particular importance to specific anatomical regions.

Conversely, we left that the statistical determination of the size and position of the R-ROIs would indicate which of the anatomical structures would assume a particular criticality for each patient. So it was not necessary to pre-process pressure data applying a mask to identify any given foot region anatomical structure [9,40,41]. In such a way, all the possible risk areas are taken into account, letting the clinician to evaluate them all and to identify each possible critical issue.

The total surfaces of R-ROIs per each foot were finally computed for comparison (see below) of the offloading insoles performances produced by the TCCI vs CAD-CAM_CMI approaches. Indeed, the optimal offloading custom-made insoles design should bring to the complete removal of over-the-risk-pressure-threshold foot areas. So, we chose to investigate the R-ROIs dimension variations as the parameter that takes into account both the over-the-

risk pressure and the size of risk regions, being the optimal custom-made insole goal the complete removal of the R-ROIs when they are present.

For clinical considerations, the correlation to the anatomical region was subsequently performed by superimposing the so determined R-ROIs pressure grids position and extent along the underfoot surface to the previously recorded and marked millimetric-blueprint feet images.

## Population sample size determination

One-way ANOVA tested the offloading performance of each design approach for repeated measures. The total area of the R-ROIs determined in three different walking conditions, namely the FI, the CAD-CAM_CMI and the TCCI orthoses, represented the dependent variable. Each foot was considered separately for its specific characteristics. Thus, the sample size has been suitably defined in terms of the total number of feet. According to the approach described by Cohen [42] and its implementation in the GPower software version 3.1.9.4 [43], we calculated that a sample size of 41 feet would provide an 80% power for the desired effect size of at least $d = 0.5$ (type I error $\alpha = 5\%$ and assuming perfect sphericity of the sample). Therefore, we chose conservatively to increase the sample size to 60 feet, thus leading to a population of 30 subjects. This choice is justified by the lack of a priori knowledge about the real value of the sphericity.

## Sample size determination (number of required steps)

A power analysis has been performed to determine the necessary number of stance phases to be recorded. Pressure values in each cell of the MPPM are compared with the set threshold of 200 kPa through One-Sample t-Test [44] to establish the R-ROIs and their areas. To this end, a prior power calculation analysis (GPower software tool version 3.1.9.4) [43], by imposing 80% test power (t-test difference from constant, one tail), type I error $\alpha = 5\%$ and desired effect size of at least $d = 0.6$, showed that a recording of at least 19 walking steps (stance phases) was needed.

Moreover, to evaluate the underfoot pressure redistribution induced by the insoles, a paired Student's t-test for each cell of the MPPM between the FI and each of the two custom-made insole design was applied. Still, the power calculation analysis (80% test power -paired t-Test two tails-, type I error $\alpha = 5\%$; effect size of at least $d = 0.6$) [43], indicated that a minimum of 24 walking steps (stance phases) recording was required. This latter test requires a more stringent condition on the number of mid-gait steps to be recorded; for this reason, we chose to apply this latter throughout this study. The decision to consider 24 walking steps brought the additional advantage to lower the effect size of One-Sample t-Test [44] from $d = 0.6$ to $d = 0.52$.

## Non-parametric statistical analysis

Since the data of the three walking conditions showed a significant deviation from normality, (Shapiro-Wilk test and QQ 'plot), the use of the non-parametric version of the ANOVA (Friedman's test) [45] has been necessary. Three Pairwise Signed-Ranks tests followed Friedman's test, and Bonferroni correction was applied (k = 3, type I error of pairwise tests $\alpha = 0.05/3$, i.e. $\alpha = 0.016667$).

In the non-parametric test, like Friedman's test, the a priori sample size computation assessed for the ANOVA One-way repeated measures case, was theoretically no more valid. A general rule of thumb states that the final sample needed can be computed by adding 15% to the required size for the corresponding parametric test [46]. Reminding that the previously determined required population sample size was n = 41, by adding the 15% it results n = 47.15 far less than the considered final sample size of n = 60.

All the statistical tests, except for the calculation of the sample size, were performed with the free tool 'Real Statistics using Excel', release 6.8 [45], available at the link: http://www.real-statistics.com/.

### Quantitative statistical framework (QSF)

Summarising, we define as the quantitative statistical framework (QSF) the full process described above, including all the following steps:

1) the acquisition of a given statistically required number of steps (at least 24 stance phases); 2) the determination of the sequence of peak pressure maps from the sequence of stance phases; 3) the computation of the MPPM; 4) the determination of R-ROIs sizes and locations through One-Sample t-Test comparison of the pressure values in each cell of the MPPM with the set threshold of 200 kPa; 5) the determination of the underfoot pressure redistribution when different insoles are donned (e.g. in the present paper FI vs custom-made insoles) through a paired Student's t-test for each cell of the MPPM; 6) the possibility to apply further statistical test such as ANOVA or non-parametric equivalent (Friedman test) to compare the performance of multiple kinds of insoles.

## Results

As compared with FI (20.6±12.9 cm$^2$), both the TCCI (7±8.7 cm$^2$) and the CAD-CAM (5.5 ±7.3 cm$^2$) approach provided a reduction of R-ROIs mean areas (p<0.0001).

Fig 4 visually shows a summarising example of the assessed R-ROIs and the related underfoot pressure redistribution induced by the insoles obtained for one subject in the three considered conditions. Precisely, the example in Fig 4, represents one of the most successful cases, for which both the TCCI and CAD-CAM_CMI approaches produced the optimal offloading removing completely the R-ROIs determined in FI condition.

Conversely, Fig 5 represents a less successful example. Even if reduced in size R-ROIs are still present in the custom-made insole condition as well, for both approaches.

By analysing the pressure redistribution surface represented by the marked white squares, it can be noted how it is possible to improve the performance of both approaches. Indeed, a large area of the midfoot, particularly under the arch, remains unchanged if compared to the FI condition. It derives that it is possible to offload R-ROIs' areas further redistributing part of their risk-pressure on the not used area under the arch.

Fig 6 reports the box plot for descriptive statistics summarising R-ROIs' areas characteristics in the three different conditions. In all cases, the custom-made insoles engendered a reduction of R-ROIs showing the beneficial effects of the personalised design. The non-parametric Friedman test (see the previous section) reported relevant significant differences among groups p = 2.9E-21. Table 2 reports posthoc pairwise signed-Ranks tests showing the better performances of CAD-CAM_CMI vs TCCI approach.

Table 3 reports mean pressure reductions (and their percentage) in residual R-ROIs for FI vs custom-made insoles. Residual R-ROIs are those risk areas that, even if reduced when compared to the initially detected R-ROIs in the FI, are not entirely removed by the custom-made offloading insole. Due to the underfoot pressure redistribution induced by custom-made offloading insoles, the residual R-ROIs are usually placed in different foot location and could present different areas as compared to those observed by using the FI. Therefore, the statistical comparison of R-ROIs average pressure values was impossible. On the other hand, we calculated the number of optimal performance cases, which indicates the number of successful complete R-ROIs removal on the considered sixty feet.

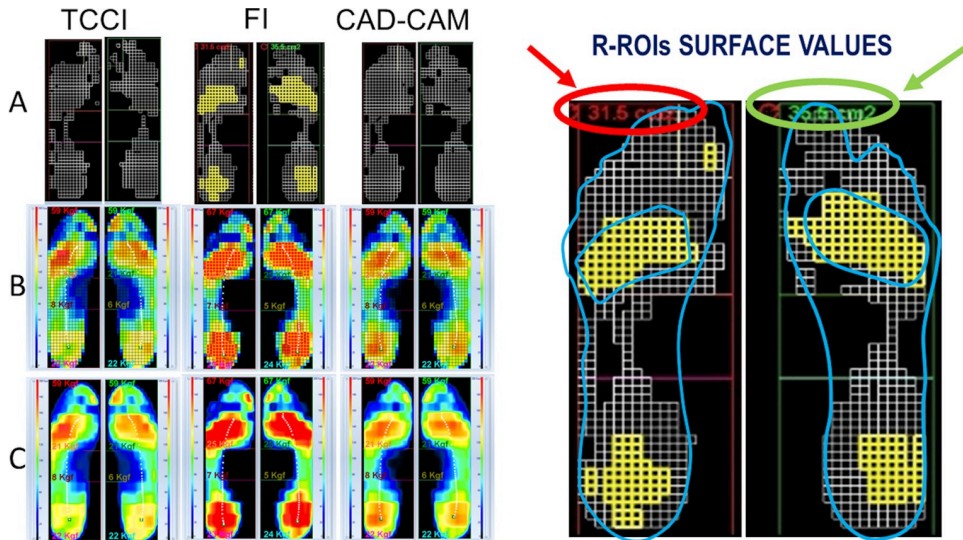

**Fig 4. Optimal offloading performance case.** Assessed R-ROIs (yellow squares) and the underfoot pressure redistribution (white squares) induced by the insoles obtained for one subject in the 3 considered conditions. The picture on the left side presents three rows, three panels each, showing the comparison for the three considered conditions, respectively. The sequence from left to right shows TCCI vs FI vs CAD-CAM_CMI, respectively. The upper row (A) shows the results of the R-ROIs identification procedure and the t-Test to detect the pressure redistribution. In the top left corner of each picture, it can be seen the computed value of the total surface (cm²) of the determined R-ROIs. The white squares have to be interpreted considering the comparing pairs FI vs TCCI and FI vs CAD-CAM_CMI representing all and only the cells statistically different between the two conditions using the paired t-Test. The panels in the middle (B) and lower (C) rows show the mean peak frames in two different graphical representations. The middle row (B) shows a 5mm x 5mm cell discretisation level. The lower row (C) shows high resolution interpolated graphical images. On the right side, the recorded blueprints with the previously marked clinical at-risk locations (i.e. callosities and sites of previous ulcers) are superimposed on the zoomed FI related images. On the top are highlighted the computed value of the R-ROIs total surface per each foot. Reprinted under a CC BY license, with permission from Bioengineering & Biomedicine Company Srl, Pescara, Italy, original copyright 2020.

It is important to note that mean pressure reductions reported in Table 3 are only related to the residual R-ROIs. I.e. the thirty-nine over sixty partial sub-optimal cases. Indeed, the real average pressure reduction is much higher when all cases are considered.

For clinical considerations, a correlation of detected R-ROIs with all the patient's underfoot specific at-risk locations (i.e. callosities and sites of previous ulcers) has been performed manually by superimposing the recorded blueprints to the so computed R-ROIs and related MPPM. See Figs 4 and 5. In all cases, a good clinical agreement has been found meaning that all R-ROIs clustered where callosities and sites of previous ulcers were marked on blueprints.

## Discussion

The aim of this study was twofold. First, we developed a sound QSF to analyse in-shoe pressure measurements. Through this method, we compared the outcomes of traditional shape-based total contact insole design with a novel CAD-CAM approach, based on a mixed 3D shape and pressure measurement information for the placement of offloading features in NDF. The results demonstrate the possibility to evaluate insoles offloading efficacy in at-risk foot regions by such a novel QSF. Moreover, we showed that both the TCCI and the CAD-CAM approach reduced R-ROIs mean areas as compared with FI. Finally, the CAD-CAM approach yielded better performance than the TCCI in terms of inducing a more significant reduction in R-ROIs sizes.

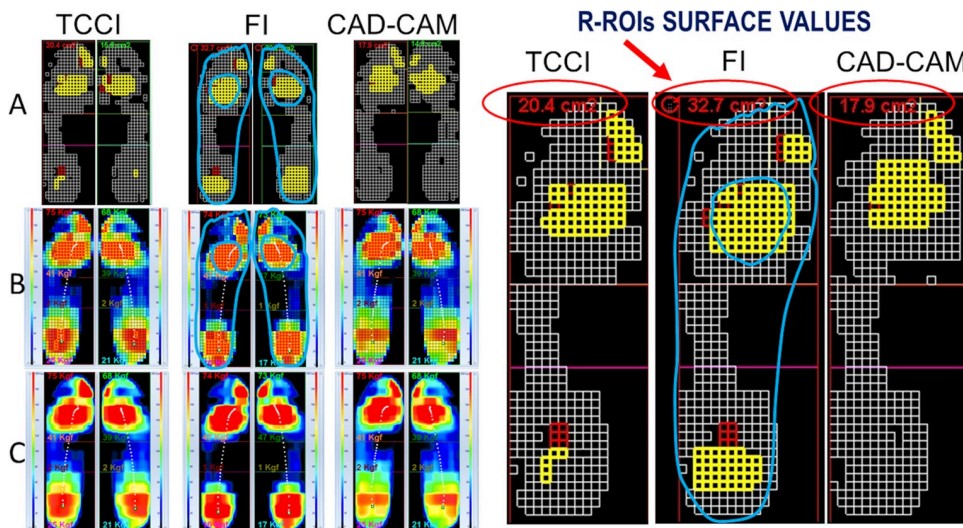

**Fig 5. Sub-optimal offloading performance case.** Assessed R-ROIs (yellow squares) and the underfoot pressure redistribution (white squares) induced by the insoles obtained for one subject in the 3 considered conditions. Red squares are introduced, and they represent those cells that belong to R-ROIs, for which no statistical difference has been detected by the paired t-Test between pairs of conditions. The sequences of the panels and their meaning are the same as in Fig 4. In this case, the custom-made insoles either TCCI or CAD-CAM_CMI were not able to remove the determined R-ROIs in the FI condition altogether (see the values in the left top corner of the panels–left foot zoomed on the right side to highlight the differences in the R-ROIs surfaces). The recorded blueprints with related previously manually marked clinical at-risk locations (i.e. callosities and sites of previous ulcers) are superimposed on the FI related images. Reprinted under a CC BY license, with permission from Bioengineering & Biomedicine Company Srl, Pescara, Italy, original copyright 2020.

Here, we confirmed that, as compared with the FI, the orthoses customising process driven by quantitative data achieves substantial offloading effects in NDF. Indeed, both the TCCI and CAD-CAM_CMI approaches engendered a significant reduction of R-ROIs as compared with the FI (Table 2 and Figs 4–6), and complete removal of R-ROIs residual was evident in twenty-one over sixty samples (35%) (Table 3), leading to optimal offloading performance in both the customised approaches. This outcome is in agreement with previous studies [13,22,37].

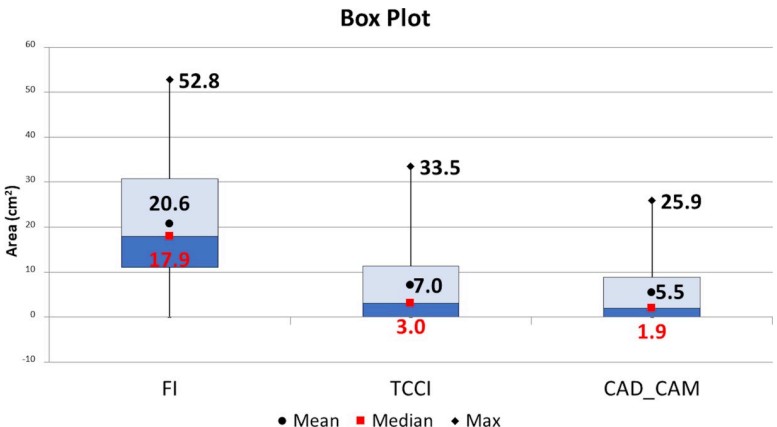

**Fig 6. Box plot for descriptive statistics summarising R-ROIs' characteristics.** Mean (circles) and Median (squares) and Max (diamonds) values are reported. All reported values are expressed in cm$^2$.

**Table 2. Post-Hoc pairwise signed-ranks test for the comparison among considered conditions.**

| Pairwise Signed-ranks tests | | | |
|---|---|---|---|
| | | p-Value | Difference of means[1] |
| FI | TCCI | 1.8E-11 | 13.64 cm$^2$ |
| FI | CAD-CAM_CMI | 1.7E-11 | 15.22 cm$^2$ |
| TCCI | CAD-CAM_CMI | 0.00083 | 1.58 cm$^2$ |

[1] The difference of means column represents the obtained reduction of R-ROIs mean areas (expressed in cm$^2$) passing from the listed condition n. 1 to the condition n.2.

The superiority of CAD-CAM quantitative strategy versus the TCCI is another important finding of our research, confirmed with a relevant statistical significance (p = 0.00083). This observation, although partly expected, further underscores the concept that in-shoe peak pressure in NDF is most effectively reduced when the orthoses customising process is driven by 3D shape and pressure quantitative information than by using the 3D shape information only. This result is in agreement with the findings of other authors [3,9,11,13].

The QSF, herein introduced, demonstrated its usefulness in identifying high-risk areas with ascertained statistical significance (i.e. R-ROIs) with detail of 5mm x 5mm pressure cells. The proposed QSF improves the direct 200kPa cut-off-based methods, allowing to properly approach and manage the intrinsic gait-related variability occurring in both the physiological and pathological gait. It is useful to emphasize that there are two possible sources of variability: the first and most intuitive is connected to the poor repeatability of each gait cycle in terms of both speed and length of the steps. The second relates to the continuous variation of the foot support modality, which is modulated, step by step, according to the balance variations and which can be particularly poorly controlled when peripheral neuropathy is present. Such variability reflects naturally on the underfoot pressure. Thus, it must be considered when the underfoot pressure measurements are used in the custom-made insole design and the monitoring of their outcome performances. Such a statistical perspective, using the average of all the peak pressure maps determined per each stance phase, provides a useful tool to avoid artefacts due to sporadic transient conditions (outliers) in the subject gait. Artefacts could lead to a misidentification of the at-risk region with the inclusion or the exclusion of not statistically validated areas. The proposed technique presents another positive feature, as it allows the possibility to determine with statistical significance how the custom-made insole redistributes the pressure along with the full underfoot surface respect to the flat insole. This information is fundamental for subsequent data-driven indications on where and how sub-optimal outcomes produced by the custom-made insole offloading design can be improved (see Fig 5). For instance, it could be possible to offload elevated pressure areas by further transferring part of the loads to areas with low underfoot pressure values where no changes are detected.

**Table 3. Mean pressure reduction in residual R-ROIs for FI vs custom-made insole comparison.**

| Mean Pressure Reduction in residual R-ROIs for FI vs Custom-made Insole Comparison | | | |
|---|---|---|---|
| | | Mean Reductions[1] non-optimal cases | N. of Optimal performance cases/60 Feet |
| FI | TCCI | 34.2 kPa (14.3%) | 21 |
| FI | CAD-CAM_CMI | 37.3 kPa (15.6%) | 21 |

[1] The Mean reductions column represents the obtained reduction in non-optimal cases, i.e. in mean pressure evaluated for the residual R-ROIs. See text for explanation.

However, some requirements must be fulfilled. In particular, using a prior power calculation analysis, we determined that at least 24 mid steps per each foot must be recorded. Indeed, such a sample size is necessary to discriminate differences of about 0.5 SD between means (i.e. the t-Test on each pressure cell for custom-made vs flat insole comparisons—Cohen's d = 0.6) or respect to a fixed threshold (i.e. the mean peak pressure cell vs the 200 kPa threshold through One-Sample t-Test—Cohen's d = 0.52) keeping the test power at 80%, and the type I error at α = 5% levels respectively. Such a requirement doubles the 12 steps limit suggested by the work of Arts et al. 2011 [41] that, if applied to our QSF, would significantly reduce the power of the test (at equal effect size). I.e. down to 47% for the t-Test comparison and 51% for the 200 kPa threshold through One-Sample t-Test comparison [42–44]. From all the above considerations, the QSF approach takes into account peak pressure variability during gait so granting from a statistical point of view a well controlled level of test power and the type I error in the identification of the risk regions. Such a framework is more rigorous than performing a direct 200kPa threshold on the average of peak pressure maps. A further advantage of the proposed QSF method is to allow the analysis of complete plantar surface areas without the need of a predetermined anatomical identification/masking. The correlation with the anatomical structure can be performed as subsequent analysis, as we showed, by superimposing the statistical outcomes to the recorded foot blueprints.

In the present paper, our aim was limited to determine if the R-ROIs positions and extents were clinically correlated to the medically evaluated at-risk-sites through blueprints recording and marking. Undoubtedly, the manual recording of blueprints introduced some human error. In any case, to perform a rigorous quantitative correlation analysis was beyond the investigation scope, as it did not influence either the introduced QSF structure or its outcomes. The rigorous quantitative correlation will be a matter of future study when appropriate digital 2D foot scanner would be used to this aim.

The monitoring along time of the correlation of detected R-ROIs with all the patient's underfoot specific clinical at-risk locations (i.e. callosities and sites of previous ulcers) can result in a better comprehension of the elevated pressure influence on either the foot ulcer formation or healing.

A recent review of Collings et al. 2020 [17] highlighted the difficulty in differentiating the effect of the different insole and footwear features in offloading the NDF. However, they found that arch profiles, metatarsal additions, and apertures are effective in reducing plantar pressure. Precisely, three meta-analyses have been presented showing peak plantar pressure mean reduction of 37 kPa when an arch profile was added, of 35.96 kPa when metatarsal additions were used, and of 75.4 kPa when pressure informed design was applied. This review recommended the use of pressure analysis to enhance the effectiveness of the design of footwear and insoles, mainly through modification. As it can be noted, the results shown in Table 3 are quite compatible with those obtained in the Collings et al. [17] meta-analysis related to arch profile support. In effect, in the present study, both the TCCI and CAD-CAM_CMI designs produce a result of arch profile support in the custom-made insole being the CAD-CAM_CMI approach more successful than TCCI by the introduction of deepened areas where R-ROIs were detected. Worth to note that the best offloading performance (mean difference of 75.4 kPa) induced by pressure informed design as described in Collings et al. [17] was derived considering data from studies where multiple subsequent modifications interventions were allowed.

Indeed, although CAD-CAM_CMI produced a better outcome in R-ROIs removal/reduction respect to TCCI, still it yielded 39 sub-optimal results out of 60 feet considered (65%). Multiple causes can be argued to explain such a sub-optimal outcome. First, as a general consideration, the information obtained through pressure and 3D shape measuring process is fundamental, but it is not sufficient to completely describe the complexity of the phenomenon

under study. Indeed, a recent study proved that using additional information, data-driven modifications improve the offloading efficacy on handmade and CAD-CAM design for patients with diabetic neuropathy at high risk of foot ulceration [25]. In particular, the study showed that the best results could be achieved even in a handmade approach if a well-structured procedure is applied. Such a procedure consisted in a detailed algorithm based on the knowledge that specifies the design elements and materials used, their hardness, thickness, and location and in-shoe plantar pressure guided insole/footwear modifications (multiple subsequent modifications are applied when pressure targets are not achieved) to further improve the insole/footwear after delivery [25]. The offloading algorithm used for the CAD-CAM insole design in this study is based on some proportional deepening under the R-ROIs areas. Such an approach could be undoubtedly improved by including computations that integrate material dynamic response characteristics. This latter, combined with novel patient-specific finite element modelling of the human foot into the algorithm [24,47] could reasonably help to advance CAD-CAM insole design. This is a matter for further study.

Indeed, aim for future study is to investigate how the R-ROIs clustering and their correlation with anatomical structures can be connected to the patient's specific foot-floor interaction and biomechanical gait characteristics. Such integration should help to identify a foot insole (and possibly footwear) design that combines not only elevated pressure zone to be removed, but also extends the evaluation of all the lower limb gait biomechanics. In this way, additional strategies can be developed to counteract possible functional-anatomical failures such as flat/arched, valgus-pronated foot, and lower limb joint problems.

Undoubtedly the literature showed that best offloading results are obtained when a well-defined protocol considering subsequent measurements/modifications is applied [11,12,14,17,18,25].

The ability of QSF to determine how the custom-made insole redistributes the pressure along the whole underfoot surface respect to the flat insole extends its usefulness beyond the evaluation step. It can provide a tool to drive the modifications necessary to remove sub-optimal offloading insole performances when included in a structured design protocol such as those already proposed in the literature [11,12,14,17,18,25].

It also represents a useful tool to monitor if the offloading performance of customised orthoses maintains their efficacy along an extended period.

A study from Waaijman et al. [11], also confirmed by Arts et al. [3], demonstrated the importance of systematic control of orthoses using in-shoe plantar pressure analysis to maintain over-time the best offloading performance. Indeed, they showed that insoles modification could be necessary over-time because of wear of material or progression of the pathology that could induce a change in the patients' gait strategies. Long term monitoring is another reason to prefer the CAD-CAM design approach. Indeed, it allows either to duplicate, quickly and accurately, the custom-made insole each time a copy is needed or to implement any required change, without starting from scratch, keeping a controlled level of accuracy.

To keep under control the statistical robustness of the study, we decided to use specific sandals able to accommodate the flat as well as the custom-made insoles. This choice allowed either to reduce the confounding effect induced by the different characteristics of the patients' shoes or the interference in the underfoot pressure measurements deriving by shoe-upper unfit.

It has been underlined that the efficacy of the designed offloading insoles could depend on the shoes adopted by the patient. Collings et al. in their review [17] wrote: "the amalgamation of features in insole and footwear designs makes consolidation of the body of knowledge difficult for understanding which features to use at which time point". The proposed QSF can be helpful to evaluate quickly such influence allowing to investigate fruitfully how the customised insole works when donned inside the patient's shoe.

Finally, we remark that the presented QSF is a very general procedure. It is not bounded to NDF investigation, but it can be fruitfully used both in clinical as well as in sport fields when foot floor biomechanical interaction and custom-made insoles design is of interest.

The study presents various limitations. Even if the sample size has been rigorously determined through statistical inference, nevertheless, the number of participants in this study represents a small sample compared to the diabetic population. For this reason, extended clinical trials have to be planned to establish the clinical relevance of the QSF method is in reducing risk of ulceration, and validate the proposed methodology as a routine clinical tool, on a larger scale.

In the results section, the data representing the mean pressure reduction induced by custom-made insole were someway incomplete. Indeed, the current algorithm version was designed to provide only pressure values for all the cells belonging to the R-ROIs, disregarding those with a pressure below the threshold (200 kPa). Due to this limitation, the pressure reduction values associated with optimal design outcome was not calculated. However, by knowing that where R-ROIs were not present, each cell shows a value significantly lower than 200kPa, the lower limit of the mean pressure reduction may be indirectly calculated by the following formula:

$$\text{Pressure Reduction} \geq \mu_P(R - ROI_{FI}) - 200\text{kPa}$$

Where $\mu_P(R\text{-}ROI_{FI})$ is the mean peak pressure value of the R-ROIs determined in the FI condition. The subsequent development of the QSF will remove such limitation.

Custom-made CAD-CAM insoles have been designed considering a single CAD-CAM tool for which partial knowledge was available to the authors. Several CAD-CAM approaches are present in the literature and the insole/footwear design market. Different algorithms and approaches could provide different outcomes, so the results regarding the better performance of CAD-CAM_CMI vs TCCI designs cannot be generalised, even if the results demonstrated that an increase of the information used in the design process reasonably leads to outcomes improvements, as it has been shown in other studies [3,11,12,14,17,18,25].

Finally, since R-ROIs were computed on the MPPM, no information was available about how long, during the stance phase, each R-ROIs element has been loaded over the 200 kPa threshold. Conversely, such a piece of information would allow better identification of the risk-level associated to each R-ROIs element. To this aim, we developed an improved version of such QSF, which is currently the object of an ongoing study.

## Conclusions

In this study, we introduced a sound QSF to analyse in-shoe pressure measurements. This framework provides a possible standardisation in the methodology of the evaluation of off-loading insoles design and subsequent monitoring steps driving to optimal offloading results guaranteeing their performance preservation along time.

Indeed, the QSF method demonstrated its usefulness in identifying high-risk areas by taking into account peak pressure variability during gait with ascertained statistical significance, a controlled level of test power and type I error in more rigorous way than performing a direct 200kPa threshold on the average of peak pressure maps. The proposed technique presents another positive feature, as it allows the possibility to determine on a statistical base the underfoot pressure redistribution induced by the custom-made foot orthosis. The evaluation of such redistribution allows the assessment of the manufactured foot orthoses performance highlighting new eventual R-ROIs to be furtherly removed. Such a process could be iterated up to the complete removal of R-ROIs. In such a way the QSF approach could demonstrate useful to help preventing the risk of ulceration.

Furthermore, our technique allows the analysis of the whole plantar surface area of the foot, with the single pressure cell detail, independently from a predetermined anatomical identification/masking.

Our data extend previous findings showing that the orthoses customising process driven by either quantitative data or clinical experience is a cornerstone in offloading design. Finally, we show that the CAD-CAM strategy achieved better offloading performance than the traditional shape-only based approach.

The use of our QSF in large-scale randomised clinical trials may thus be warranted to ascertain better the outcomes (ulcer prevention, recurrence, limb amputation) in different at-risk diabetic patients.

The possibility to use QSF to provide a detailed description of how and where custom-made insole redistributes the pressure along with the whole underfoot surface respect to the flat insole extends its usefulness to the design step. Indeed, it can help to guide the modifications necessary to achieve optimal offloading insole performances.

Such a QSF is a very general procedure, its application extends beyond diabetic foot investigation, being potentially useful every time foot-floor biomechanical interaction and custom-made insoles design is of interest as in other clinical as well as in sport fields.

## Supporting information

**S1 File. R-ROIs data set for FI vs TCCI and FI vs CAD-CAM_CMI comparisons.** (XLSX)

## Acknowledgments

The authors acknowledge the company: Guantificio Altotiberino Ecosanit Calzature Snc, for providing laboratory access, the measurement device and access to the software for data elaboration and CAD-CAM design, the APC fees. The funders had no role in the design of the study; in the collection, analyses, or interpretation of data; in the writing of the manuscript, or in the decision to publish the results.

## Author Contributions

**Conceptualization:** Moreno D'Amico, Piero Roncoletta.

**Data curation:** Moreno D'Amico, Edyta Kinel, Piero Roncoletta, Andrea Gnaldi, Celeste Ceppitelli, Federico Belli, Giuseppe Murdolo, Cristiana Vermigli.

**Formal analysis:** Moreno D'Amico, Edyta Kinel, Piero Roncoletta, Cristiana Vermigli.

**Investigation:** Moreno D'Amico, Edyta Kinel, Piero Roncoletta, Andrea Gnaldi, Celeste Ceppitelli, Federico Belli, Giuseppe Murdolo, Cristiana Vermigli.

**Methodology:** Moreno D'Amico, Piero Roncoletta, Cristiana Vermigli.

**Project administration:** Moreno D'Amico.

**Resources:** Moreno D'Amico, Piero Roncoletta.

**Software:** Moreno D'Amico, Piero Roncoletta, Andrea Gnaldi.

**Supervision:** Moreno D'Amico, Edyta Kinel, Piero Roncoletta, Cristiana Vermigli.

**Validation:** Moreno D'Amico, Edyta Kinel, Piero Roncoletta, Andrea Gnaldi, Celeste Ceppitelli, Federico Belli, Giuseppe Murdolo, Cristiana Vermigli.

**Visualization:** Moreno D'Amico, Edyta Kinel, Piero Roncoletta, Andrea Gnaldi, Celeste Ceppitelli, Federico Belli, Giuseppe Murdolo, Cristiana Vermigli.

**Writing – original draft:** Moreno D'Amico, Edyta Kinel, Piero Roncoletta, Andrea Gnaldi, Celeste Ceppitelli, Federico Belli, Giuseppe Murdolo, Cristiana Vermigli.

**Writing – review & editing:** Moreno D'Amico, Edyta Kinel, Piero Roncoletta, Cristiana Vermigli.

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
