## [Decision Letter · Decision Letter 0]

17 Jul 2020

PONE-D-20-14175

Data-driven CAD-CAM vs traditional total contact custom insoles: a novel quantitative-statistical framework for the evaluation of insoles offloading performance in diabetic foot

PLOS ONE

Dear Dr. D'Amico,

Thank you for submitting your manuscript to PLOS ONE. After careful consideration, we feel that it has merit but does not fully meet PLOS ONE’s publication criteria as it currently stands. Therefore, we invite you to submit a revised version of the manuscript that addresses the points raised during the review process.

Although the reviewers impressions have been quite positive, they do address some major components requiring changes. Please respond to their comments in detail. 

We look forward to receiving your revised manuscript.

Kind regards,

Kevin Mattheus Moerman, Ph.D.

Academic Editor

PLOS ONE

Journal Requirements:

2. We note that Figure(s) [#] in your submission contain copyrighted images. All PLOS content is published under the Creative Commons Attribution License (CC BY 4.0), which means that the manuscript, images, and Supporting Information files will be freely available online, and any third party is permitted to access, download, copy, distribute, and use these materials in any way, even commercially, with proper attribution. For more information, see our copyright guidelines: http://journals.plos.org/plosone/s/licenses-and-copyright.

1.    You may seek permission from the original copyright holder of Figure(s) [#] to publish the content specifically under the CC BY 4.0 license.

Reviewers' comments:

Reviewer's Responses to Questions

**Comments to the Author**

1. Is the manuscript technically sound, and do the data support the conclusions?

Reviewer #1: Partly

Reviewer #2: Yes

Reviewer #3: Yes

Reviewer #4: Yes

2. Has the statistical analysis been performed appropriately and rigorously? 

Reviewer #1: Yes

Reviewer #2: Yes

Reviewer #3: Yes

Reviewer #4: Yes

3. Have the authors made all data underlying the findings in their manuscript fully available?

Reviewer #1: Yes

Reviewer #2: Yes

Reviewer #3: Yes

Reviewer #4: Yes

4. Is the manuscript presented in an intelligible fashion and written in standard English?

Reviewer #1: Yes

Reviewer #2: Yes

Reviewer #3: Yes

Reviewer #4: Yes

5. Review Comments to the Author

Reviewer #1: Thank you for your publication and contribution to an area that is lacking robust evidence.

Overall, I am struggling to understand the quantitative statistical framework -this needs to be more explicit within your article. What is it, what was the method, make it more explicit in the results section.

line 27 throughout the article you interchange use the terms insoles and orthotics and sometimes add in footwear - can you be consistent and use one term throughout (eg line 29)

line 50 - neuropathy and neuropathic in same sentence

line 55 add in 'the' before foot

line 55 very long sentence - split into 2

line 60 add a full stop

line 61 footwear is also an important aspect of reducing foot pressures and usually works in conjunction with the insole....especially as you refer to outer sole in line 74

ln 85 add 'an' at start of sentence

ln 85/86 - this sentence does not make sense

ln 87 remove also

ln 88 clarify 'periodic systematic control'

ln 90 remove 'still'

ln 91 remove nevertheless

ln 95 remove up

ln 99 write info in full

ln 109 - was acute vascular problems an exclusion?

ln 121 - peripheral neuropathy consists of motor, sensory and autonomic - do you mean sensory in this sentence?

ln 133 - who was double blinded?

ln 133 clarify subsequent

ln 134 special sandals requires better description and company

ln 135 neutral flat insole - why do you need to use neutral? - flat insole would suffice but requires more description eg materials, pictures

ln 136 - how much time between 1st and 2nd sessions?

were the sandals worn just for the testing or also between sessions by participants?

ln 137-141 - rewrite to make more readable

ln 143 - clarify all foot othoses are TCCI and cad cam (flat insole is also a device)

ln 143 and ln 151 contradict as you say eva used only and then you add a top cover

ln 169 - clarify if TCCI and Cad cam orthoses; incidentally who made the flat insoles?

ln 182- reword for clarification (eg each participant recieved ...)

ln 186 - clarify if experimenatl session 1 or 2

ln 186 clarify if the two testing conditions are TCCI and cad cam

ln 206 experimental sandals

ln 225 - how did the speed of testing session 1 compare with session 2?

ln 238 - clarify why not necessary to use masking to reader

the parametric and non-parametric analysis - please clarify the purpose of these tests (eg we used the test to determine....)

ln 331 - you say descrptive but i cannot find them

ln 338 - you are not examining the distribution of roi's - reword

ln 341 - define beneficial

percentage reduction in pressure would be a useful addition to your absolute data as it would account for the large variations in participants barefoot pressure data that absolute values do not show

ln 389 - robust? i am not clear how it is robust as you have not presented any justification for this

ln 396 better performance - in terms of what?

ln 399 - valuable point but not appropriate in this paragrapgh

ln 411 - 'perfect' is a very strong word to use in this context

ln 418 - what tests did you use to justify its reliability? i don't think you did so can't claim this

ln 447-448 i am still not sure what the QSF framework is

ln 452 - what about different gait strategies and changes over time?

ln 455 full stop missing

Reviewer #2: The paper presents a comparative study between three different types of orthotics in order to evaluate which is the best for reducing the foot plantar pressure. The evaluation is conducted considering 30 patients and different production technologies.

The manuscript is well structured and clear in its contents. The analysis methodology is presented in detail. The conclusions are supported by well-analyzed clinical data.

Custom made CAD-CAM insoles have been designed considering a single CAD tool. For the reviewer, this is the main limitation of the work. The effectiveness of an insole is strictly dependent on its geometry and therefore on the algorithm/software used to define its geometry. The reviewer recommends adding information regarding the offloading algorithm used to generate the 3D geometry of orthotics, as well as the CAD modelling procedure. As a future development, it would be interesting to see a comparison between orthotics developed with different modelling software

Reviewer #3: This is an interesting topic of investigation, as we are in need of effective offloading insoles for the diabetic foot. The authors introduce a new framework to analyze pressure pictures and obtain results from, but I am trying to find the real benefits of this approach over regular in-shoe peak pressure analysis from the pressure distribution pictures, which is quite a straightforward analysis itself. What is the problem we have that stimulates the development of this new framework? How is it going to help us better to interpret in-shoe pressure pictures?

Specific comments

Abstract

- Line 33: this is not a double-blind controlled trial. It is a cross-sectional study design

- Line 36: very few quantitative results are reported compared to lots of text in the abstract, which is a bit odd given the QSF, a quantitative method that the authors are using

Introduction

- In line 63, the authors refer to poor adherence being a barrier to clinical success with insole designs. This is supported by trial data from Bus et al.1, an important reference that is missing from the reference list.

- In line 76 the authors state: “Plantar pressure can be redistributed but not eradicated and reducing stress at one location may simply displace the risk of ulceration to a different area of the foot.” This is true for the average pressure obtained, but not necessarily for the peak of the plantar pressure, a parameter that is mostly reported in the diabetic foot literature. I would suggest to specify this in the paper.

- In line 91 the authors state: Nevertheless, the strategies to design the optimal offloading insoles have to be fully standardised. In the absence of such standardisation, outcomes for digital quantitative data-driven approaches can be debatable in comparison of traditional foot orthoses supply chains [14].” Recently, two papers have been published, specifically addressing this topic, which stress that standardization and data-drive design protocols are needed, although not necessarily via CADCAM design principles2,3

- In line 95, the authors state their aim to: “Firstly, we wanted to build up a robust quantitative-statistical framework (QSF) to analyse in-shoe pressure measurements to propose a way to standardise the evaluation of offloading insoles' design and subsequent monitoring steps.” But in the section before the aim, the authors state that the design of insoles needs to be standardized. So, is it the evaluation of the insole design or the design itself that has to be standardized, as these are two different things? If referring to the design, this is more valuable than when referring to the evaluation.

Methods

- Why were mostly risk class 1 subjects recruited, as the insoles studies would normally apply more to a risk class 2 or 3 patient?

- Line 171: “Plantar pressure data, which were integrated into its algorithms for the manufacture of the custom insoles, were also provided.” Which plantar pressure data is meant? More detail is required here

- Line 182: why was a double-blind design considered important for this study? If not blinded, how would that have affected the results?

- Line 234: Why were “The total surfaces of R-ROIs per each foot“ chosen as the primary variable for comparison between offloading insoles, and why not the peak pressure, which is the most commonly reported parameter. And what was the rationale for using a parameter like R-ROI and not analyzing the pressure values in a more straightforward fashion as done by others?

- Line 241 onwards: the authors present the study, as the title shows as a “a novel quantitative-statistical framework”. It is not clear what is novel about it and what the ‘statistical framework’ is, as it seems that straightforward parametric and non-parametric tests are used, comparing the different conditions in a cross-sectional study fashion.

- Line 257: why is for the power calculation of “underfoot pressure redistribution” the number of steps indicated, whereas normally, and for the primary parameters of R-ROI, the number of patients is reported?

Results

The results section is difficult to read and requires improvement

- Line 279 The authors use a subheading, but this is the only one in the results section, so redundant

- Line 293 to 306 reads as a legend of the Figure and should be moved to the legend, so that the results section only contains the results of the analyses

- The same for lines 319 to 325

- Lines 358 – 380 contain a lot of explanation rather than results, that one would expect in the methods or discussion section

- There are 8 figures and 3 tables, which is a lot for a study using just 2 parameters, straightforward analyses and very few quantitative results. In particular, the results on ROI-R are sometimes difficult to follow

- I don’t see any reference in the results to the regions that are most important to offload in the neuropathic diabetic patient, like the hallux, 1st metatarsal head, where foot ulcers most commonly occur. How does the framework take that into account, as some regions are more important to analyze than others? And how does it do that in an efficient way, as from peak pressure distribution pictures from the in-shoe pressure measurement, this information can be directly observed.

Discussion

- Line 389 – 405 read as an introduction. I suggest to discuss the results of the study sooner

- Line 418. The authors state that “ The quantitative-statistical framework herein introduced demonstrated its reliability and usefulness in identifying high-risk areas with ascertained statistical significance (i.e. R-ROIs), representing a marked improvement compared to the methods based on the direct 200kPa cut-off.” Where in the results can I find results on the reliability and usefulness of the framework and how robust it is, as the authors state in the conclusions? And based on which outcome do they conclude that this is a marked improvement compared to using the 200 kPa cutoff method?

- Line 438: “ Long term monitoring is another reason to prefer the CAD-CAM design approach….”. This is contrary to a recent study that shows that handmade custom-made insoles provide more pressure relief than CADCAM insoles.3

References

1. Bus SA, Waaijman R, Arts M, et al. Effect of custom-made footwear on foot ulcer recurrence in diabetes: a multicenter randomized controlled trial. Diabetes Care 2013; 36(12): 4109-16.

2. Bus SA, Zwaferink JB, Dahmen R, Busch-Westbroek T. State of the art design protocol for custom made footwear for people with diabetes and peripheral neuropathy. Diabetes Metab Res Rev 2020; 36 Suppl 1: e3237.

3. Zwaferink JBJ, Custers W, Paardekooper I, Berendsen HA, Bus SA. Optimizing footwear for the diabetic foot: Data-driven custom-made footwear concepts and their effect on pressure relief to prevent diabetic foot ulceration. PLoS One 2020; 15(4): e0224010.

Reviewer #4: Data-driven CAD-CAM vs traditional total contact custom insoles: a novel quantitative statistical

framework for the evaluation of insoles offloading performance in diabetic foot.

This study, in the field of the diabetic foot and neuropathic ulcer prevention, is important as it incorporates and develops the use of computer-aided design and manufacturing in the quantitative evaluation and design of custom-made orthotics for plantar pressure relief, in direct comparison to that for traditional total contact cast insoles.

The unique point of this study is the enhanced use of digital technology in this field, which is highly likely to be the way forward for any future standardised design of custom-made insoles for optimal pressure off-loading.

Major points:

Overall, sections could be shortened. Certain paragraphs would benefit from being made less wordy.

Could the authors comment on their finding that the TCCI and CAD-CAM_CMI approaches did not produce good offloading in some cases (Fig. 5). What was unique about these individuals? Was it related to their specific characteristics such as foot deformities, gait etc? Or is this a methodology problem?

The methodology for obtaining the ‘blueprint’ images of the feet on a millimetric grid has not been detailed, however, I presume from Fig. 2 that the method involves manually drawing around the foot, then estimating the ‘areas of concern’ by observing an equivalent photo of the foot, then manually drawing the ROI’s onto the blueprint. This manual approach is not a standardised process and will undoubtedly introduce human error. The blueprint image and photo image in Fig. 2 are not even the same size for direct comparison. This is a study limitation and should be discussed. The authors may wish to draw comparisons to a recently published method which uses technology to standardise diabetic foot plantar images, developed to avoid such human error (eg. Yap et al, JDST, 2018).

The Results section is too long. Eight figures and 3 tables is very excessive. My initial thoughts are that Figs. 4 & 5 could be combined into one figure (a & b), as could Figs. 7 and 8. Data from tables 2 and 3 could be incorporated into the text.

There is no numerical data shown to back up the claims made about the statistical correlation of R-ROIs and at-risk locations (lines 383-387).

Minor points:

‘In-shoe dynamic planar pressure measurement’ (or something similar) may be a better sub-heading than ‘Instrumentation’.

There is repetition of methodology in lines 188-189 again at line 211-213.

Some sweeping statements in the Discussion should be re-phrased, to better indicate that this study’s results merely confirm or reflect results from other studies.

e.g. line 411 “This outcome is in perfect agreement with previous studies.”

line 416 “This result is in full agreement with the findings of other authors.”

6. PLOS authors have the option to publish the peer review history of their article (what does this mean?). If published, this will include your full peer review and any attached files.

Reviewer #1: **Yes: **Richard Collings

Reviewer #2: No

Reviewer #3: No

Reviewer #4: No

---

## [Author Response · Author response to Decision Letter 0]

21 Aug 2020

ANSWERS TO REVIEWERS

Please find hereinafter our detailed answer to each point raised by the academic editor and reviewer(s)

PREMISE

We want to thank the reviewers for their suggestions, comments and critics that resulted helpful and useful to stimulate us to improve the submitted paper. 

We were working hard to rewrite essential paragraphs and sections to fulfil their suggestions and to explicit all the underlying concepts we left as “tacit” and “implicit” in the previous version.

Indeed, after receiving comments and suggestions from the reviewers, we realised that the main reason that led us to develop the proposed quantitative statistical framework, i.e. the necessity to approach and manage the “Gait natural variability”, was not given as an explicit motivation. For such a reason we modified the manuscript in the Introduction, Methods, Results, and Discussion sections to introduce and discuss at length such a concept.

We do believe that now the final revision of the paper is more understandable and more comprehensive. Each point raised by the academic editor and reviewer(s) has been considered in the revision. 

In the following, we provide a specific answer to all the raised points:

N.B. Please note all the given line numbers in the following are referred to the unmarked version of the manuscript.

Reviewer #1: Thank you for your publication and contribution to an area that is lacking robust evidence.

Overall, I am struggling to understand the quantitative statistical framework -this needs to be more explicit within your article. What is it, what was the method, make it more explicit in the results section.

We thank the reviewer #1 for such a comment.

We realised that the main reason that led us to create the quantitative statistical framework (QSF), i.e. the creation of a method to manage the natural gait variability appropriately was not openly declared. So, we modified the Introduction section where we raised such an issue explaining why such an issue has to be approached. 

Please refer to lines 104-114 

Then we added in the Methods section a full paragraph to clearly describe what the QSF is, its meaning, and all the requirements to properly manage data analysis. 

Please refer to lines 340-350

In the Results section, the importance of QSF is straightforward.

Please refer to lines 353-444

In the Discussion section, we discussed at length how the proposed QSF is the adequate necessary tools to handle the quantitative underfoot pressure measurements, considering the intrinsic gait variability, showing its superiority to the direct 200kPa threshold.

Please refer to lines 466-493

line 27 throughout the article you interchange use the terms insoles and orthotics and sometimes add in footwear - can you be consistent and use one term throughout (eg line 29)

We thank the reviewer #1 for such a comment. 

We recognised that this could be an issue. To be as much general as possible, because several healthcare disciplines may be involved in the provision of footwear and insoles for people with diabetes, we referred to the definitions proposed in van Netten et al., where the terms “custom-made insoles” and “custom-made in-shoe orthosis/orthotic” are equivalent and in a such a way they will be used throughout this paper.

Please refer to lines 69-74

line 50 - neuropathy and neuropathic in same sentence 

Removed

line 55 add in ‘the’ before foot

Done

line 55 very long sentence - split into 2

Done

line 60 add a full stop

Done

line 61 footwear is also an important aspect of reducing foot pressures and usually works in conjunction with the insole....especially as you refer to outer sole in line 74

Added in the Introduction a clarification of the footwear role

Please refer to lines 65-68

We also added in the Discussion section further considerations about the footwear role and the difficulty in differentiating the effect of the different insole and footwear features in offloading the neuropathic diabetic foot as found in the very recent review published by reviewer#1. We found this recent review very useful to strengthen some of the concepts we wanted to stress in our paper.

Please refer to lines 507-521 and 568-573

ln 85 add ‘an’ at start of sentence

Sentence removed

ln 85/86 - this sentence does not make sense

Sentence removed

ln 87 remove also

Done

ln 88 clarify ‘periodic systematic control’

Added in the Introduction a clarification.

Please refer to lines 97-99

ln 90 remove ‘still’

Done

ln 91 remove nevertheless

Done

ln 95 remove up

Done

ln 99 write info in full

Done

Ln 109 - was acute vascular problems an exclusion?

Thank you for noticing.

YES. We added in the exclusion criteria

Please refer to line 136

ln 121 - peripheral neuropathy consists of motor, sensory and autonomic - do you mean sensory in this sentence?

Added sensory

ln 133 - who was double blinded?

A clarification was added in the Methods.

Please refer to lines 159-161

ln 133 clarify subsequent

Removed

ln 134 special sandals requires better description and company

A clarification was added in the Methods.

Please refer to lines 155-156

ln 135 neutral flat insole - why do you need to use neutral? - flat insole would suffice but requires more description eg materials, pictures

The word neutral was removed throughout the paper

A clarification was added in the Methods section for flat insole material and manufacturer.

Please refer to lines 169-172

ln 136 - how much time between 1st and 2nd sessions?

A clarification was added in the Methods.

Please refer to lines 213-217

were the sandals worn just for the testing or also between sessions by participants?

A clarification was added in the Methods.

Please refer to line 217

ln 137-141 - rewrite to make more readable

Rewritten in the Methods.

Please refer to lines 161-167

ln 143 - clarify all foot othoses are TCCI and cad cam (flat insole is also a device)

Please see above

ln 143 and ln 151 contradict as you say eva used only and then you add a top cover

Rewritten in the Methods.

Please refer to lines 169-171

ln 169 - clarify if TCCI and Cad cam orthoses; incidentally who made the flat insoles?

Clarified.

Please refer to line 197

For flat insole manufacturer, please read above

ln 182- reword for clarification (eg each participant recieved ...)

Rewritten in the Methods.

Please refer to lines 213-227

ln 186 - clarify if experimenatl session 1 or 2

Rewritten in the Methods.

Please refer to lines 213-227

ln 186 clarify if the two testing conditions are TCCI and cad cam

Rewritten in the Methods.

Please refer to lines 213-227

ln 206 experimental sandals

Added the word “experimental”.

Please refer to line 244

ln 225 - how did the speed of testing session 1 compare with session 2?

A clarification was added in the “In-Shoe Dynamic Plantar Pressure Measurement” sub-section.

Please refer to lines 245-252

ln 238 - clarify why not necessary to use masking to reader

Rewritten in the Data Analysis.

Please refer to lines 287-290

the parametric and non-parametric analysis - please clarify the purpose of these tests (eg we used the test to determine....)

Rewritten completely by introducing three specific sub-sections

Population sample size determination

Sample size determination (number of required steps)

Non-Parametric Statistical analysis.

Please refer to lines 300-339

ln 331 - you say descrptive but i cannot find them

The whole sentence was removed (it was a typing error left by chance)

ln 338 - you are not examining the distribution of roi’s - reword

Rewritten in the Results.

Word “distribution” removed.

Please refer to line 402

ln 341 - define beneficial

Rewritten in the Results.

Please refer to lines 403-404

percentage reduction in pressure would be a useful addition to your absolute data as it would account for the large variations in participants barefoot pressure data that absolute values do not show

Added in Table 3

Please refer to lines 431-435

ln 389 - robust? i am not clear how it is robust as you have not presented any justification for this

Reworded as “sound”.

ln 396 better performance - in terms of what?

Rewritten in the Discussion.

Please refer to lines 452-453

ln 399 - valuable point but not appropriate in this paragrapgh

Removed

ln 411 - ‘perfect’ is a very strong word to use in this context

Removed

ln 418 - what tests did you use to justify its reliability? i don’t think you did so can’t claim this

Rewritten in the Discussion the word “reliability” was removed.

Please refer to lines 466-467

ln 447-448 i am still not sure what the QSF framework is

Please see above Introduction, Methods, Results, Discussion.

ln 452 - what about different gait strategies and changes over time?

We explained throughout the paper that QSF method allows precisely to evaluate the influence of the different walking strategies as well as to monitor their changes over time

ln 455 full stop missing

Done

Reviewer #2: The paper presents a comparative study between three different types of orthotics in order to evaluate which is the best for reducing the foot plantar pressure. The evaluation is conducted considering 30 patients and different production technologies.

The manuscript is well structured and clear in its contents. The analysis methodology is presented in detail. The conclusions are supported by well-analysed clinical data.

Custom made CAD-CAM insoles have been designed considering a single CAD tool. For the reviewer, this is the main limitation of the work. The effectiveness of an insole is strictly dependent on its geometry and therefore on the algorithm/software used to define its geometry. The reviewer recommends adding information regarding the offloading algorithm used to generate the 3D geometry of orthotics, as well as the CAD modelling procedure. As a future development, it would be interesting to see a comparison between orthotics developed with different modelling software

We thank the reviewer #2 for her/his comments. 

We added information about the CAD modelling procedure that was not disclosed to the authors because of Intellectual Property reasons

Please refer to lines 206-209, where we added:

“The automated design algorithm details are the intellectual property of Guantificio Altotiberino ECOSANIT Calzature Snc, Anghiari, Italy, and they were not disclosed to the authors being not strictly necessary for the present study.”

We also added specific considerations in the limitations of the study paragraph.

Please refer to lines 591-597

Reviewer #3: This is an interesting topic of investigation, as we are in need of effective offloading insoles for the diabetic foot. The authors introduce a new framework to analyse pressure pictures and obtain results from, but I am trying to find the real benefits of this approach over regular in-shoe peak pressure analysis from the pressure distribution pictures, which is quite a straightforward analysis itself. What is the problem we have that stimulates the development of this new framework? How is it going to help us better to interpret in-shoe pressure pictures?

We thank the reviewer #3 for her/his comments. 

Please, refer to the above-written PREMISE.

Refer as well to modifications we introduced in the paper. 

We explained in various sections (Introduction, Methods, Results, Discussion) the motivation which led us to propose the QSF. 

We clarified why QSF provides better solution respect to direct 200kPa threshold methods proposed in the literature

Specific comments

Abstract

- Line 33: this is not a double-blind controlled trial. It is a cross-sectional study design

Removed double-blinded in the abstract and added explanations in the Methods section.

Please refer to lines 159-161

- Line 36: very few quantitative results are reported compared to lots of text in the abstract, which is a bit odd given the QSF, a quantitative method that the authors are using

Added some more numerical results considering the Abstract length limit of 300 words.

Introduction

- In line 63, the authors refer to poor adherence being a barrier to clinical success with insole designs. This is supported by trial data from Bus et al.1, an important reference that is missing from the reference list.

Added in the reference list

- In line 76 the authors state: “Plantar pressure can be redistributed but not eradicated and reducing stress at one location may simply displace the risk of ulceration to a different area of the foot.” This is true for the average pressure obtained, but not necessarily for the peak of the plantar pressure, a parameter that is mostly reported in the diabetic foot literature. I would suggest to specify this in the paper.

We are not sure to understand the following part of the comment from a theoretical point of view

“but not necessarily for the peak of the plantar pressure, a parameter that is mostly reported in the diabetic foot literature. I would suggest to specify this in the paper”

In any case from our results, it has been demonstrated that the mean peak pressure maps were indeed modified by custom-made insoles respect to the flat insole. Potentially a wrong offloading modification, in the custom-made insoles, can displace R-ROIs (i.e. cells clusters that resulted, with ascertained statistical significance, pressure values, higher than 200kPa risk of ulcer) from one specific underfoot region to a different one. 

- In line 91 the authors state: Nevertheless, the strategies to design the optimal offloading insoles have to be fully standardised. In the absence of such standardisation, outcomes for digital quantitative data-driven approaches can be debatable in comparison of traditional foot orthoses supply chains [14].” Recently, two papers have been published, specifically addressing this topic, which stress that standardisation and data-drive design protocols are needed, although not necessarily via CADCAM design principles2,3

Added in the reference list

- In line 95, the authors state their aim to: “Firstly, we wanted to build up a robust quantitative-statistical framework (QSF) to analyse in-shoe pressure measurements to propose a way to standardise the evaluation of offloading insoles’ design and subsequent monitoring steps.” But in the section before the aim, the authors state that the design of insoles needs to be standardised. So, is it the evaluation of the insole design or the design itself that has to be standardised, as these are two different things? If referring to the design, this is more valuable than when referring to the evaluation.

We clarified in the Methods and the Discussion, how QSF could help in the process of standardisation of both evaluation and design protocols.

Please refer to lines 340-350 (Evaluation)

Please refer to lines 475-482 and 549-563 (Design)

Methods

- Why were mostly risk class 1 subjects recruited, as the insoles studies would normally apply more to a risk class 2 or 3 patient?

As explained in the Methods, 220 patients were screened to obtain the needed 30 (number of patients determined through statistical inference see Population sample size determination sub-section) patients population satisfying the inclusion/exclusion criteria. Once the necessary number of patients was reached, the research could start.

- Line 171: “Plantar pressure data, which were integrated into its algorithms for the manufacture of the custom insoles, were also provided.” Which plantar pressure data is meant? More detail is required here

Added and clarified in the Methods section

Please refer to lines 199-203

- Line 182: why was a double-blind design considered important for this study? If not blinded, how would that have affected the results?

Added and clarified in the Methods section

Please refer to lines 159-161

- Line 234: Why were “The total surfaces of R-ROIs per each foot“ chosen as the primary variable for comparison between offloading insoles, and why not the peak pressure, which is the most commonly reported parameter. And what was the rationale for using a parameter like R-ROI and not analysing the pressure values in a more straightforward fashion as done by others?

Please, refer to the above-written PREMISE.

Refer as well to modifications we introduced in the paper. 

We explained in various sections the motivation which led us to propose the QSF. 

We clarified why QSF provides better solution respect to direct 200kPa threshold methods proposed in the literature

The R-ROIs (i.e. cells clusters that resulted, with ascertained statistical significance, pressure values higher than 200kPa risk of ulcer) are implicitly considering the pressure values as explained. Their sizes provide quantitative information on how much the offloading modifications in the custom-made insole resulted successfully.

- Line 241 onwards: the authors present the study, as the title shows as a “a novel quantitative-statistical framework”. It is not clear what is novel about it and what the ‘statistical framework’ is, as it seems that straightforward parametric and non-parametric tests are used, comparing the different conditions in a cross-sectional study fashion.

Please, refer to the above-written PREMISE.

Refer as well to modifications we introduced in the paper. 

We explained in various sections the motivation which led us to propose the QSF. 

We clarified why QSF provides better solution respect to direct 200kPa threshold methods proposed in the literature

Besides, we added in the Methods section, a full paragraph to clearly describe what the QSF is, its meaning, and all the requirements to properly manage data analysis. 

Please refer to lines 340-350

Furthermore, sub-sections in the Methods have been rewritten entirely to explain better the statistical test applied. In particular, three specific sub-sections have been introduced:

Population sample size determination

Sample size determination (number of required steps)

Non-Parametric Statistical analysis.

Please refer to lines 300-339

- Line 257: why is for the power calculation of “underfoot pressure redistribution” the number of steps indicated, whereas normally, and for the primary parameters of R-ROI, the number of patients is reported?

Sub-sections in the Methods have been rewritten entirely to explain better the statistical test applied, In particular, three specific sub-sections have been introduced:

Population sample size determination

Sample size determination (number of required steps)

Non-Parametric Statistical analysis.

Please refer to lines 300-339

Results

The results section is difficult to read and requires improvement

Results section has been fully rewritten 

- Line 279 The authors use a subheading, but this is the only one in the results section, so redundant

Removed

- Line 293 to 306 reads as a legend of the Figure and should be moved to the legend, so that the results section only contains the results of the analyses

- The same for lines 319 to 325

Moved into the legends

- Lines 358 – 380 contain a lot of explanation rather than results, that one would expect in the methods or discussion section

Partially moved to the Discussion section and partially rewritten.

Please refer to lines 581-590

- There are 8 figures and 3 tables, which is a lot for a study using just 2 parameters, straightforward analyses and very few quantitative results. In particular, the results on ROI-R are sometimes difficult to follow

Figure 7 and 8 have been removed

- I don’t see any reference in the results to the regions that are most important to offload in the neuropathic diabetic patient, like the hallux, 1st metatarsal head, where foot ulcers most commonly occur. How does the framework take that into account, as some regions are more important to analyse than others? And how does it do that in an efficient way, as from peak pressure distribution pictures from the in-shoe pressure measurement, this information can be directly observed.

Please reread the revised version of the paper. We are confident now after such revision that everything should be clear. In any case, by seeing the Results section, either in Figure 4 or Figure 5, the R-ROIs are precisely indicating that the patients considered in such examples were presenting at-risk areas under the metatarsal heads and halluces. 

Discussion

- Line 389 – 405 read as an introduction. I suggest to discuss the results of the study sooner

They were partially removed. Discussion section rewritten; results discussed sooner.

- Line 418. The authors state that “ The quantitative-statistical framework herein introduced demonstrated its reliability and usefulness in identifying high-risk areas with ascertained statistical significance (i.e. R-ROIs), representing a marked improvement compared to the methods based on the direct 200kPa cut-off.” Where in the results can I find results on the reliability and usefulness of the framework and how robust it is, as the authors state in the conclusions? And based on which outcome do they conclude that this is a marked improvement compared to using the 200 kPa cutoff method?

Please reread the revised version of the paper. We are confident now after such revision that everything should be clear.

The word “reliability” has been removed not to generate confusion with “statistical reliability.”

Please refer to line 466

- Line 438: “ Long term monitoring is another reason to prefer the CAD-CAM design approach….”. This is contrary to a recent study that shows that handmade custom-made insoles provide more pressure relief than CADCAM insoles.3

We are not sure to understand the above comment, in that it does not appear in any contrast with the cited paper 3. Moreover, such a study proved that considering additional information, data-driven modifications improve the offloading efficacy on handmade and CAD-CAM design as well.

In any case, we added further considerations, including this latter in the Discussion section.

Please refer to lines 524-534

References

1. Bus SA, Waaijman R, Arts M, et al. Effect of custom-made footwear on foot ulcer recurrence in diabetes: a multicenter randomised controlled trial. Diabetes Care 2013; 36(12): 4109-16.

2. Bus SA, Zwaferink JB, Dahmen R, Busch-Westbroek T. State of the art design protocol for custom made footwear for people with diabetes and peripheral neuropathy. Diabetes Metab Res Rev 2020; 36 Suppl 1: e3237.

3. Zwaferink JBJ, Custers W, Paardekooper I, Berendsen HA, Bus SA. Optimising footwear for the diabetic foot: Data-driven custom-made footwear concepts and their effect on pressure relief to prevent diabetic foot ulceration. PLoS One 2020; 15(4): e0224010.

All the three references have been added in the reference list

Reviewer #4: Data-driven CAD-CAM vs traditional total contact custom insoles: a novel quantitative statistical

framework for the evaluation of insoles offloading performance in diabetic foot.

This study, in the field of the diabetic foot and neuropathic ulcer prevention, is important as it incorporates and develops the use of computer-aided design and manufacturing in the quantitative evaluation and design of custom-made orthotics for plantar pressure relief, in direct comparison to that for traditional total contact cast insoles.

The unique point of this study is the enhanced use of digital technology in this field, which is highly likely to be the way forward for any future standardised design of custom-made insoles for optimal pressure offloading.

We thank the reviewer #4 for her/his comments. 

Major points:

Overall, sections could be shortened. Certain paragraphs would benefit from being made less wordy.

The paper has been intensively reworked following the reviewers’ suggestions and requests. We tried to shorten as much as possible the sections but unfortunately given the number of issues to be approached under the suggestions and requests of the reviewers it was not possible to fulfil such a suggestion truly. 

Could the authors comment on their finding that the TCCI and CAD-CAM_CMI approaches did not produce good offloading in some cases (Fig. 5). What was unique about these individuals? Was it related to their specific characteristics such as foot deformities, gait etc? Or is this a methodology problem?

We added specific considerations in the Discussion section.

Please refer to lines 522-563

The methodology for obtaining the ‘blueprint’ images of the feet on a millimetric grid has not been detailed, however, I presume from Fig. 2 that the method involves manually drawing around the foot, then estimating the ‘areas of concern’ by observing an equivalent photo of the foot, then manually drawing the ROI’s onto the blueprint. This manual approach is not a standardised process and will undoubtedly introduce human error. The blueprint image and photo image in Fig. 2 are not even the same size for direct comparison. This is a study limitation and should be discussed. The authors may wish to draw comparisons to a recently published method which uses technology to standardise diabetic foot plantar images, developed to avoid such human error (eg. Yap et al, JDST, 2018).

We added specific considerations in the Discussion section.

Please refer to lines 495-503

The Results section is too long. Eight figures and 3 tables is very excessive. My initial thoughts are that Figs. 4 & 5 could be combined into one figure (a & b), as could Figs. 7 and 8. Data from tables 2 and 3 could be incorporated into the text.

Figure 7 and 8 have been removed and the text shortened. It has been impossible to combine the figures and the tables as suggested without compromising the readability of the paper and presented data. 

There is no numerical data shown to back up the claims made about the statistical correlation of R-ROIs and at-risk locations (lines 383-387).

We added specific considerations in the Discussion section.

We claimed just that the correlation between at-risk areas marked on the blueprints and statistically determined R-ROIs was satisfying from a clinical point of view.

Please refer to lines 495-503

Minor points:

‘In-shoe dynamic planar pressure measurement’ (or something similar) may be a better sub-heading than ‘Instrumentation’.

Changed in “In-shoe dynamic plantar pressure measurement. “

There is repetition of methodology in lines 188-189 again at line 211-213.

Removed

Some sweeping statements in the Discussion should be re-phrased, to better indicate that this study’s results merely confirm or reflect results from other studies.

e.g. line 411 “This outcome is in perfect agreement with previous studies.”

line 416 “This result is in full agreement with the findings of other authors.”

The words: “perfect” and “full” were Removed

---

## [Decision Letter · Decision Letter 1]

27 Oct 2020

PONE-D-20-14175R1

Data-driven CAD-CAM vs traditional total contact custom insoles: a novel quantitative-statistical framework for the evaluation of insoles offloading performance in diabetic foot

PLOS ONE

Dear Dr. D'Amico,

Thank you for submitting your manuscript to PLOS ONE. After careful consideration, we feel that it has merit but does not fully meet PLOS ONE’s publication criteria as it currently stands. Therefore, we invite you to submit a revised version of the manuscript that addresses the points raised during the review process.

Although reviewers 1 and 2 are happy to accept this submission, reviewers 3 and 4 have remaining concerns which have not been adequately addressed. Please accurately deal with their comments and appropriately respond to their queries in a clear and concise manor. One reviewer expresses concerns about the length of the article. Since other reviewers have, in this review process, asked for additional content it may be natural and acceptable that the article has grown in length. However I do urge you to consider this reviewer's concerns and aim at formulating sections in a more concise fashion where possible.

We look forward to receiving your revised manuscript.

Kind regards,

Kevin Mattheus Moerman, Ph.D.

Academic Editor

PLOS ONE

Reviewers' comments:

Reviewer's Responses to Questions

**Comments to the Author**

1. If the authors have adequately addressed your comments raised in a previous round of review and you feel that this manuscript is now acceptable for publication, you may indicate that here to bypass the “Comments to the Author” section, enter your conflict of interest statement in the “Confidential to Editor” section, and submit your "Accept" recommendation.

Reviewer #1: All comments have been addressed

Reviewer #2: All comments have been addressed

Reviewer #3: (No Response)

Reviewer #4: (No Response)

2. Is the manuscript technically sound, and do the data support the conclusions?

Reviewer #1: Yes

Reviewer #2: Yes

Reviewer #3: Partly

Reviewer #4: Yes

3. Has the statistical analysis been performed appropriately and rigorously? 

Reviewer #1: Yes

Reviewer #2: Yes

Reviewer #3: I Don't Know

Reviewer #4: Yes

4. Have the authors made all data underlying the findings in their manuscript fully available?

Reviewer #1: Yes

Reviewer #2: Yes

Reviewer #3: Yes

Reviewer #4: Yes

5. Is the manuscript presented in an intelligible fashion and written in standard English?

Reviewer #1: Yes

Reviewer #2: Yes

Reviewer #3: No

Reviewer #4: Yes

6. Review Comments to the Author

Reviewer #1: thank you for your revisions. This is an excellent article that contributes to the offloading evidence for DFU.

Reviewer #2: (No Response)

Reviewer #3: Comments

The authors have done a large effort to revise the manuscript based on the reviewer’s comments, improving clarity. However, they do not always adequately respond to questions raised by the reviewers and often merely refer to the revised manuscript, where a response to the point raised is expected.

The authors introduce gait variability as argument to develop the QSF, but what is the clinical rationale for this? How does variability in pressure affect foot ulceration, suggesting that we need better methods that take into account these variabilities?

Line 284 – 286: “Such intrinsic variability connected to gait, both normal and pathological one, implies the necessity to approach the study of its characteristics from a statistical point of view. That is, by defining a rigorous averaging process to extract mean behaviours and their associated variability from which to derive clinically relevant parameters with a statistical significance.”

Ok, but to what extent do the authors fulfill this requirement?

Line 300-303: “We underline that such statistical R-ROIs determination permits to manage for the intrinsic gait variability considering both the mean value and its related SD. In such a way, the drawback of a simple direct threshold comparison performed on a single peak frame map is overcome by making the determination of R-ROIs position and extent a more theoretically rigorous process.”

The authors imply that the 200 kPa threshold method uses a single peak pressure frame map of the measurement of multiple step in-shoe pressure in the patient. This is not the case. A mean peak pressure map over a minimum 12 steps of in-shoe plantar pressure is used to identify regions with peak pressure >200 kPa and therefore also accounts for variability in pressure between steps, as reflected by the mean peak pressure over these steps. This should be corrected in the manuscript

What the 200 kPa threshold method does not do is use the SD of the mean peak pressure, but from the paper it is not clear yet to me in what exact way the QSF does take the SD into account and why that is important beyond from a theoretical point of view. How does this work exactly?

The design process of the CADCAM insoles is quite similar to Owing et al.(ref 9), but there is no reference to their approach in the methods section. This should be added

Line 307-308. “For such a reason we did not give any a priori particular importance to specific anatomical regions.” Why not? Since specific regions are much more susceptible for ulceration than other. For example, in-shoe pressures in the heel are often among the highest found in the foot, but ulcers hardly ever occur on the plantar heel, so less important for the analysis and outcome

Based on their outline of comparisons between the QSF and the >200 kPa threshold method, their statement that “From all the above considerations, the direct 200kPa threshold appears theoretically

inadequate to grant the appropriate at-risk regions identification.” is too bold. See above comments.

Furthermore, The 200kPa level method has been proven very useful in optimizing diabetic footwear and has also been identified as an appropriate threshold below which foot ulcers can be prevented when the footwear is worn (ref 41 in their paper). The QSF has not yet shown clinical relevance, merely a biomechanical or theoretical one, and since the aim of custom-made footwear is to reduce risk of ulceration, the authors should be careful with stating that demonstrated clinically relevant methods are inadequate. At most, the author may present the QSF as an alternative method that takes variability in peak pressure more into account in a statistical way and that it remains to be proven what the clinical relevance of this method is in reducing risk of ulceration

Conclusions:

Line 703: “Indeed, the method demonstrated its usefulness in identifying high-risk areas with ascertained statistical significance (R-ROIs) improving the direct 200kPa cut-off approach outlined in the literature.” Should be modified based on above argumentation

Line 705: “The proposed technique presents another positive feature, as it allows the possibility to

determine on a statistical base the underfoot pressure redistribution induced by the custom-made

foot orthosis, thus preventing the risk of ulceration in different underfoot regions.”

This reads as if the technique itself reduces the risk of ulceration, where the authors mean to say I think that with the technique better offloading insoles can be designed that may reduce risk of ulceration. This should be better reflected in the conclusions

Authors should check their references: some papers seem to appear twice in the reference list

Authors should also check grammar and typo's and English writing in the paper

Reviewer #4: I requested that manuscript sections could be shortened. However, I find that many of the new phrases added to the various sections are over-elaborate, sometimes a little confusing to read, and have added considerably to the word count.

I asked a specific question regarding why the TCCI and CAD-CAM-CMI approaches did not

produce good offloading in some cases, and speculated if it was related to individuals’ specific characteristics such as foot deformities, gait etc? Or a methodology problem? The authors referred me to a new, 40-line paragraph in the Discussion; however, this does not appear to answer my specific question.

My other queries have been answered satisfactorily.

7. PLOS authors have the option to publish the peer review history of their article (what does this mean?). If published, this will include your full peer review and any attached files.

Reviewer #1: **Yes: **Richard Collings

Reviewer #2: No

Reviewer #3: No

Reviewer #4: No

---

## [Author Response · Author response to Decision Letter 1]

13 Nov 2020

PLOS ONE Decision: Revision required [PONE-D-20-14175R1] - [EMID:abb0860dbd965f93]

Posta in arrivo

PLOS ONE <em@editorialmanager.com>

27 ott 2020, 10:43

a me

CC: kevin.moerman@gmail.com, kevin.moerman@nuigalway.ie

PONE-D-20-14175R1

Data-driven CAD-CAM vs traditional total contact custom insoles: a novel quantitative-statistical framework for the evaluation of insoles offloading performance in diabetic foot

PLOS ONE

Dear Dr. D'Amico,

Thank you for submitting your manuscript to PLOS ONE. After careful consideration, we feel that it has merit but does not fully meet PLOS ONE’s publication criteria as it currently stands. Therefore, we invite you to submit a revised version of the manuscript that addresses the points raised during the review process.

Although reviewers 1 and 2 are happy to accept this submission, reviewers 3 and 4 have remaining concerns which have not been adequately addressed. Please accurately deal with their comments and appropriately respond to their queries in a clear and concise manor. One reviewer expresses concerns about the length of the article. Since other reviewers have, in this review process, asked for additional content it may be natural and acceptable that the article has grown in length. However I do urge you to consider this reviewer's concerns and aim at formulating sections in a more concise fashion where possible.

We look forward to receiving your revised manuscript.

Kind regards,

Kevin Mattheus Moerman, Ph.D.

Academic Editor

PLOS ONE

Reviewers' comments:

Reviewer's Responses to Questions

Comments to the Author

1. If the authors have adequately addressed your comments raised in a previous round of review and you feel that this manuscript is now acceptable for publication, you may indicate that here to bypass the “Comments to the Author” section, enter your conflict of interest statement in the “Confidential to Editor” section, and submit your "Accept" recommendation.

Reviewer #1: All comments have been addressed

Reviewer #2: All comments have been addressed

Reviewer #3: (No Response)

Reviewer #4: (No Response)

2. Is the manuscript technically sound, and do the data support the conclusions?

Reviewer #1: Yes

Reviewer #2: Yes

Reviewer #3: Partly

Reviewer #4: Yes

3. Has the statistical analysis been performed appropriately and rigorously?

Reviewer #1: Yes

Reviewer #2: Yes

Reviewer #3: I Don't Know

Reviewer #4: Yes

4. Have the authors made all data underlying the findings in their manuscript fully available?

Reviewer #1: Yes

Reviewer #2: Yes

Reviewer #3: Yes

Reviewer #4: Yes

5. Is the manuscript presented in an intelligible fashion and written in standard English?

Reviewer #1: Yes

Reviewer #2: Yes

Reviewer #3: No

Reviewer #4: Yes

6. Review Comments to the Author

Reviewer #1: thank you for your revisions. This is an excellent article that contributes to the offloading evidence for DFU.

Reviewer #2: (No Response)

Reviewer #3: Comments

The authors have done a large effort to revise the manuscript based on the reviewer’s comments, improving clarity. However, they do not always adequately respond to questions raised by the reviewers and often merely refer to the revised manuscript, where a response to the point raised is expected.

The authors introduce gait variability as argument to develop the QSF, but what is the clinical rationale for this? How does variability in pressure affect foot ulceration, suggesting that we need better methods that take into account these variabilities?

Line 284 – 286: “Such intrinsic variability connected to gait, both normal and pathological one, implies the necessity to approach the study of its characteristics from a statistical point of view. That is, by defining a rigorous averaging process to extract mean behaviours and their associated variability from which to derive clinically relevant parameters with a statistical significance.”

Ok, but to what extent do the authors fulfill this requirement?

Line 300-303: “We underline that such statistical R-ROIs determination permits to manage for the intrinsic gait variability considering both the mean value and its related SD. In such a way, the drawback of a simple direct threshold comparison performed on a single peak frame map is overcome by making the determination of R-ROIs position and extent a more theoretically rigorous process.”

The authors imply that the 200 kPa threshold method uses a single peak pressure frame map of the measurement of multiple step in-shoe pressure in the patient. This is not the case. A mean peak pressure map over a minimum 12 steps of in-shoe plantar pressure is used to identify regions with peak pressure >200 kPa and therefore also accounts for variability in pressure between steps, as reflected by the mean peak pressure over these steps. This should be corrected in the manuscript

What the 200 kPa threshold method does not do is use the SD of the mean peak pressure, but from the paper it is not clear yet to me in what exact way the QSF does take the SD into account and why that is important beyond from a theoretical point of view. How does this work exactly?

ANSWER

We thank the reviewer for the above group of questions posed. Being them basically all connected to each other we think is better to give a comprehensive answer for all of them. 

We think that the clinical rationale for the introduction of such complex QSF is aimed to have the most conservative as possible approach in considering any possible area of risk. This is accomplished only if gait variability is considered in its complete expression.

It is true, as the reviewer points out, that variability information is partially considered in the computed mean peak pressure map. But processing this information through a simple direct comparison to the 200kPa threshold could lead to neglect some of the possible risk areas, as the standard deviation information is not considered, for its statistical contribution, as is the case in the QSF approach.

To better explain such a concept, we included a simple numerical example. So, we modified/added to the manuscript the following: (lines 282-290 of the marked manuscript) 

“In such a way, the drawback of a simple direct threshold comparison performed on a mean peak frame map is overcome by making the determination of R-ROIs position and extent a theoretically more rigorous process. Indeed, a series of cells having average values fairly below the 200 kPa threshold but with high SD (let us say µ=170kPa±30kPa where SD=30kPa) would definitely not be considered by the direct threshold method. Conversely, if the variability of gait were found to be high producing a high SD (as in the numerical example), the one-sample t-test (included in the QSF) would likely consider this series of cells not statistically different from the 200 kPa threshold. Thus, the potential risk associated with this series of cells is not neglected by including them in the R-ROIs. For these reasons, the QSF approach is more conservative considering the statistical risk.”

We also added in the Discussion section (lines 486-490 of the marked manuscript) the following considerations about the possible sources of variability that can affect the underfoot pressure:

“It is useful to emphasize that there are two possible sources of variability: the first and most intuitive is connected to the poor repeatability of each gait cycle in terms of both speed and length of the steps. The second relates to the continuous variation of the foot support modality, which is modulated, step by step, according to the balance variations and which can be particularly poorly controlled when peripheral neuropathy is present.”

In addition, about the number of steps to be used to calculate the mean peak pressure map, the work of Arts et al. 2011 [41] suggest a minimum of 12 steps. However, “a mean peak pressure map over a minimum 12 steps of in-shoe plantar pressure used to identify regions with peak pressure >200 kPa” provides inadequate statistical power when the described QSF is used. We described already such situation also from a numerical point of view in the Discussion section (lines 503-512 of the marked manuscript) that we report also here for clarity:

“However, some requirements must be fulfilled. In particular, using a prior power calculation analysis, we determined that at least 24 mid steps per each foot must be recorded. Indeed, such a sample size is necessary to discriminate differences of about 0.5 SD between means (i.e. the t-Test on each pressure cell for custom-made vs flat insole comparisons - Cohen’s d=0.6) or respect to a fixed threshold (i.e. the mean peak pressure cell vs the 200 kPa threshold through One-Sample t-Test - Cohen’s d=0.52) keeping the test power at 80%, and the type I error at α=5% levels respectively. Such a requirement doubles the 12 steps limit suggested by the work of Arts et al. 2011 [41] that, if applied to our QSF, would significantly reduce the power of the test (at equal effect size). I.e. down to 47% for the t-Test comparison and 51% for the 200 kPa threshold through One-Sample t-Test comparison [42–44]”

The design process of the CADCAM insoles is quite similar to Owing et al.(ref 9), but there is no reference to their approach in the methods section. This should be added

ANSWER

We thank the reviewer for such a note. We added the following sentences at lines 199-202 of the marked manuscript:

“Such a method is similar to that presented by Owings et al. [9]. The main substantial differences, in the actual CAD-CAM_CMI design approach, are related to the use of baropodometric insoles to determine the mean pressure maps driving the insole design and to the statistically rigorously determined number of necessary steps to be averaged (see below).”

Line 307-308. “For such a reason we did not give any a priori particular importance to specific anatomical regions.” Why not? Since specific regions are much more susceptible for ulceration than other. For example, in-shoe pressures in the heel are often among the highest found in the foot, but ulcers hardly ever occur on the plantar heel, so less important for the analysis and outcome

ANSWER

We thank the reviewer for the questions. We clarified the concept lengthening the explanation in the Data Analysis section (lines 294-300 of the marked manuscript):

“For such a reason we did not give any a priori particular importance to specific anatomical regions. Conversely, we left that the statistical determination of the size and position of the R-ROIs would indicate which of the anatomical structures would assume a particular criticality for each patient. So it was not necessary to pre-process pressure data applying a mask to identify any given foot region anatomical structure [9,40,41]. In such a way, all the possible risk areas are taken into account, letting the clinician to evaluate them all and to identify each possible critical issue.”

Furthermore, the independence of the QSF from the predetermined specific anatomical regions pre-processing reveals to be useful to analyse the underfoot pressure redistribution induced by the custom-made foot orthoses as remarked in the Discussion section (lines 640-648 of the marked manuscript):

“The proposed technique presents another positive feature, as it allows the possibility to determine on a statistical base the underfoot pressure redistribution induced by the custom-made foot orthosis. The evaluation of such redistribution allows the assessment of the manufactured foot orthoses performance highlighting new eventual R-ROIs to be furtherly removed. Such a process could be iterated up to the complete removal of R-ROIs. In such a way the QSF approach could demonstrate useful to help preventing the risk of ulceration. Furthermore, our technique allows the analysis of the whole plantar surface area of the foot, with the single pressure cell detail, independently from a predetermined anatomical identification/masking.”

Based on their outline of comparisons between the QSF and the >200 kPa threshold method, their statement that “From all the above considerations, the direct 200kPa threshold appears theoretically

inadequate to grant the appropriate at-risk regions identification.” is too bold. See above comments.

ANSWER

We thank the reviewer for such a note. We modified the text as the following (lines 512-515 of the marked manuscript):

“From all the above considerations, the QSF approach takes into account peak pressure variability during gait so granting from a statistical point of view a well controlled level of test power and the type I error in the identification of the risk regions. Such a framework is more rigorous than performing a direct 200kPa threshold on the average of peak pressure maps.”

Furthermore, The 200kPa level method has been proven very useful in optimizing diabetic footwear and has also been identified as an appropriate threshold below which foot ulcers can be prevented when the footwear is worn (ref 41 in their paper). The QSF has not yet shown clinical relevance, merely a biomechanical or theoretical one, and since the aim of custom-made footwear is to reduce risk of ulceration, the authors should be careful with stating that demonstrated clinically relevant methods are inadequate. At most, the author may present the QSF as an alternative method that takes variability in peak pressure more into account in a statistical way and that it remains to be proven what the clinical relevance of this method is in reducing risk of ulceration

ANSWER

We thank the reviewer for such a note. We modified the text in the Discussion section as the following (lines 606-608 of the marked manuscript):

“For this reason, extended clinical trials have to be planned to establish the clinical relevance of the QSF method is in reducing risk of ulceration, and validate the proposed methodology as a routine clinical tool, on a larger scale”.

Conclusions:

Line 703: “Indeed, the method demonstrated its usefulness in identifying high-risk areas with ascertained statistical significance (R-ROIs) improving the direct 200kPa cut-off approach outlined in the literature.” 

Should be modified based on above argumentation

ANSWER

We thank the reviewer for such a note. We modified the text in the Conclusions section as the following (lines 637-640 of the marked manuscript):

“Indeed, the QSF method demonstrated its usefulness in identifying high-risk areas by taking into account peak pressure variability during gait with ascertained statistical significance, a controlled level of test power and type I error in more rigorous way than performing a direct 200kPa threshold on the average of peak pressure maps.”

Line 705: “The proposed technique presents another positive feature, as it allows the possibility to

determine on a statistical base the underfoot pressure redistribution induced by the custom-made

foot orthosis, thus preventing the risk of ulceration in different underfoot regions.”

This reads as if the technique itself reduces the risk of ulceration, where the authors mean to say I think that with the technique better offloading insoles can be designed that may reduce risk of ulceration. This should be better reflected in the conclusions

ANSWER

We thank the reviewer for such a note. We modified the text in the Conclusions section as the following (lines 640-645 of the marked manuscript):

“The proposed technique presents another positive feature, as it allows the possibility to determine on a statistical base the underfoot pressure redistribution induced by the custom-made foot orthosis. The evaluation of such redistribution allows the assessment of the manufactured foot orthoses performance highlighting new eventual R-ROIs to be furtherly removed. Such a process could be iterated up to the complete removal of R-ROIs. In such a way the QSF approach could demonstrate useful to help preventing the risk of ulceration.”

Authors should check their references: some papers seem to appear twice in the reference list

ANSWER

We thank the reviewer for such a note. We removed duplications from the References.

Authors should also check grammar and typo's and English writing in the paper

ANSWER

We thank the reviewer for such a note. We reprocessed all the manuscript in multiple ways also using specific software for language check.

Reviewer #4: I requested that manuscript sections could be shortened. However, I find that many of the new phrases added to the various sections are over-elaborate, sometimes a little confusing to read, and have added considerably to the word count.

ANSWER

We thank the reviewer for such a note. Indeed, we agree and we tried to shrink the text as much as possible, but to answer to all the questions posed by the reviewers made to shorten the text a real hard task.

I asked a specific question regarding why the TCCI and CAD-CAM-CMI approaches did not

produce good offloading in some cases, and speculated if it was related to individuals’ specific characteristics such as foot deformities, gait etc? Or a methodology problem? The authors referred me to a new, 40-line paragraph in the Discussion; however, this does not appear to answer my specific question.

ANSWER

We thank the reviewer for such a note. We are sorry in effect we answered to the question, it is a complex question that needs to be answered at length with various considerations. Unluckily the answer was included in the manuscript at different lines we previously indicated. We are sorry for the error.

The answer is contained in the Discussion section at lines 549-566 of actual marked manuscript. We report here the extract to avoid any further error.

“Indeed, although CAD-CAM_CMI produced a better outcome in R-ROIs removal/reduction respect to TCCI, still it yielded 39 sub-optimal results out of 60 feet considered (65%). Multiple causes can be argued to explain such a sub-optimal outcome. First, as a general consideration, the information obtained through pressure and 3D shape measuring process is fundamental, but it is not sufficient to completely describe the complexity of the phenomenon under study. Indeed, a recent study proved that using additional information, data-driven modifications improve the offloading efficacy on handmade and CAD-CAM design for patients with diabetic neuropathy at high risk of foot ulceration [25]. In particular, the study showed that the best results could be achieved even in a handmade approach if a well-structured procedure is applied. Such a procedure consisted in a detailed algorithm based on the knowledge that specifies the design elements and materials used, their hardness, thickness, and location and in-shoe plantar pressure guided insole/footwear modifications (multiple subsequent modifications are applied when pressure targets are not achieved) to further improve the insole/footwear after delivery [25]. The offloading algorithm used for the CAD-CAM insole design in this study is based on some proportional deepening under the R-ROIs areas. Such an approach could be undoubtedly improved by including computations that integrate material dynamic response characteristics. This latter, combined with novel patient-specific finite element modelling of the human foot into the algorithm [24,47] could reasonably help to advance CAD-CAM insole design. This is a matter for further study.

My other queries have been answered satisfactorily.

7. PLOS authors have the option to publish the peer review history of their article (what does this mean?). If published, this will include your full peer review and any attached files.

Do you want your identity to be public for this peer review? For information about this choice, including consent withdrawal, please see our Privacy Policy.

Reviewer #1: Yes: Richard Collings

Reviewer #2: No

Reviewer #3: No

Reviewer #4: No

---

## [Decision Letter · Decision Letter 2]

4 Feb 2021

PONE-D-20-14175R2

Data-driven CAD-CAM vs traditional total contact custom insoles: a novel quantitative-statistical framework for the evaluation of insoles offloading performance in diabetic foot

PLOS ONE

Dear Dr. D'Amico,

Thank you for submitting your manuscript to PLOS ONE. After careful consideration, we feel that it has merit but does not fully meet PLOS ONE’s publication criteria as it currently stands. Therefore, we invite you to submit a revised version of the manuscript that addresses the points raised during the review process.

My verdict at the moment is "minor revision". Please carefully address the remaining issues raised by the reviewer and resubmit.

We look forward to receiving your revised manuscript.

Kind regards,

Kevin Mattheus Moerman, Ph.D.

Academic Editor

PLOS ONE

Reviewers' comments:

Reviewer's Responses to Questions

**Comments to the Author**

1. If the authors have adequately addressed your comments raised in a previous round of review and you feel that this manuscript is now acceptable for publication, you may indicate that here to bypass the “Comments to the Author” section, enter your conflict of interest statement in the “Confidential to Editor” section, and submit your "Accept" recommendation.

Reviewer #3: All comments have been addressed

Reviewer #4: All comments have been addressed

2. Is the manuscript technically sound, and do the data support the conclusions?

Reviewer #3: Yes

Reviewer #4: Yes

3. Has the statistical analysis been performed appropriately and rigorously? 

Reviewer #3: Yes

Reviewer #4: Yes

4. Have the authors made all data underlying the findings in their manuscript fully available?

Reviewer #3: Yes

Reviewer #4: Yes

5. Is the manuscript presented in an intelligible fashion and written in standard English?

Reviewer #3: Yes

Reviewer #4: Yes

6. Review Comments to the Author

Reviewer #3: The authors have made a signifiant effort to address additional comments to the manuscript

The only aspects that needs modifications is the comparison between the QSF and the 200 kPa method in the conclusions of the abstract, that should read in a similar way as amended in the discussion and conclusion sections of the manuscript

"The introduced QSF improves the direct 200kPa cut-off approach outlined in the literature." is not correct as there is no clinical validation of the QSF in the paper, and there is for the 200 kPa method

The sentence should read as: "The introduced QSF provides a more rigorous method to the direct 200kPa cut-off approach outlined in the literature."

Reviewer #4: (No Response)

7. PLOS authors have the option to publish the peer review history of their article (what does this mean?). If published, this will include your full peer review and any attached files.

Reviewer #3: No

Reviewer #4: No

---

## [Author Response · Author response to Decision Letter 2]

5 Feb 2021

PONE-D-20-14175R2

Data-driven CAD-CAM vs traditional total contact custom insoles: a novel quantitative-statistical framework for the evaluation of insoles offloading performance in diabetic foot

PLOS ONE

Dear Dr. D'Amico,

Thank you for submitting your manuscript to PLOS ONE. After careful consideration, we feel that it has merit but does not fully meet PLOS ONE’s publication criteria as it currently stands. Therefore, we invite you to submit a revised version of the manuscript that addresses the points raised during the review process.

My verdict at the moment is "minor revision". Please carefully address the remaining issues raised by the reviewer and resubmit.

We look forward to receiving your revised manuscript.

Kind regards,

Kevin Mattheus Moerman, Ph.D.

Academic Editor

PLOS ONE

Reviewers' comments:

Reviewer's Responses to Questions

Comments to the Author

1. If the authors have adequately addressed your comments raised in a previous round of review and you feel that this manuscript is now acceptable for publication, you may indicate that here to bypass the “Comments to the Author” section, enter your conflict of interest statement in the “Confidential to Editor” section, and submit your "Accept" recommendation.

Reviewer #3: All comments have been addressed

Reviewer #4: All comments have been addressed

2. Is the manuscript technically sound, and do the data support the conclusions?

Reviewer #3: Yes

Reviewer #4: Yes

3. Has the statistical analysis been performed appropriately and rigorously?

Reviewer #3: Yes

Reviewer #4: Yes

4. Have the authors made all data underlying the findings in their manuscript fully available?

Reviewer #3: Yes

Reviewer #4: Yes

5. Is the manuscript presented in an intelligible fashion and written in standard English?

Reviewer #3: Yes

Reviewer #4: Yes

6. Review Comments to the Author

Reviewer #3: The authors have made a signifiant effort to address additional comments to the manuscript

The only aspects that needs modifications is the comparison between the QSF and the 200 kPa method in the conclusions of the abstract, that should read in a similar way as amended in the discussion and conclusion sections of the manuscript

"The introduced QSF improves the direct 200kPa cut-off approach outlined in the literature." is not correct as there is no clinical validation of the QSF in the paper, and there is for the 200 kPa method

The sentence should read as: "The introduced QSF provides a more rigorous method to the direct 200kPa cut-off approach outlined in the literature."

ANSWER

We thank the reviewer for such a note. We modified the text as the following (lines 41-42 of the marked manuscript):

“The introduced QSF provides a more rigorous method to the direct 200kPa cut-off approach outlined in the literature.”

Reviewer #4: (No Response)

7. PLOS authors have the option to publish the peer review history of their article (what does this mean?). If published, this will include your full peer review and any attached files.

Do you want your identity to be public for this peer review? For information about this choice, including consent withdrawal, please see our Privacy Policy.

Reviewer #3: No

Reviewer #4: No

---

## [Editor Report · Decision Letter 3]

17 Feb 2021

Data-driven CAD-CAM vs traditional total contact custom insoles: a novel quantitative-statistical framework for the evaluation of insoles offloading performance in diabetic foot

PONE-D-20-14175R3

Dear Dr. D'Amico,

We’re pleased to inform you that your manuscript has been judged scientifically suitable for publication and will be formally accepted for publication once it meets all outstanding technical requirements.

Kind regards,

Kevin Mattheus Moerman, Ph.D.

Academic Editor

PLOS ONE
---

## [Editor Report · Acceptance letter]

22 Feb 2021

PONE-D-20-14175R3 

Data-driven CAD-CAM vs traditional total contact custom insoles: a novel quantitative-statistical framework for the evaluation of insoles offloading performance in diabetic foot 

Dear Dr. D’Amico:

I'm pleased to inform you that your manuscript has been deemed suitable for publication in PLOS ONE. Congratulations! Your manuscript is now with our production department. 

Kind regards, 

on behalf of

Dr. Kevin Mattheus Moerman 

Academic Editor

PLOS ONE